# VGLUT1 functions as a glutamate/proton exchanger with chloride channel activity in hippocampal glutamatergic synapses

Magalie Martineau [1,4], Raul E. Guzman[2], Christoph Fahlke[2] & Jürgen Klingauf[1,3]

Glutamate is the major excitatory transmitter in the vertebrate nervous system. To maintain synaptic efficacy, recycling synaptic vesicles (SV) are refilled with glutamate by vesicular glutamate transporters (VGLUTs). The dynamics and mechanism of glutamate uptake in intact neurons are still largely unknown. Here, we show by live-cell imaging with pH- and chloride-sensitive fluorescent probes in cultured hippocampal neurons of wild-type and VGLUT1-deficient mice that in SVs VGLUT functions as a glutamate/proton exchanger associated with a channel-like chloride conductance. After endocytosis most internalized $Cl^-$ is substituted by glutamate in an electrically, and presumably osmotically, neutral manner, and this process is driven by both the $Cl^-$ gradient itself and the proton motive force provided by the vacuolar $H^+$-ATPase. Our results shed light on the transport mechanism of VGLUT under physiological conditions and provide a framework for how modulation of glutamate transport via $Cl^-$ and pH can change synaptic strength.

[1] Department of Cellular Biophysics, Institute for Medical Physics and Biophysics, University of Muenster, 48149 Muenster, Germany. [2] Institute of Complex Systems, Zelluläre Biophysik (ICS-4), Forschungszentrum Jülich, 52425 Jülich, Germany. [3] IZKF Münster and Cluster of Excellence EXC 1003, Cells in Motion (CiM), 48149 Muenster, Germany. [4] Present address: University of Bordeaux and Centre National de la Recherche Scientifique, Interdisciplinary Institute for Neuroscience, UMR 5297, F-33000 Bordeaux, France. Correspondence and requests for materials should be addressed to
M.M. (email: magalie.martineau@u-bordeaux.fr) or to J.K. (email: klingauf@uni-muenster.de)

D uring neurotransmission, synaptic vesicles (SV) are retrieved by endocytosis and refilled with neuro-transmitters for a new round of exocytosis[1,2]. The filling of SV with glutamate depends on the activity of vesicular gluta-mate transporters (VGLUTs). Three isoforms of mammalian VGLUTs (VGLUT1-3) have been identified[3]. Loss of VGLUT abolishes glutamatergic neurotransmission leading to severe cognitive malfunctions and lethality[4–6]. While VGLUT is essen-tial for normal synaptic function[3], the modulation of its activity or expression is implicated in the pathophysiology of several neurological and psychiatric diseases including schizophrenia[7,8], Alzheimer's disease[9], Parkinson's disease[10,11] and epilepsy[12,13]. When heterologously expressed in Xenopus oocytes, VGLUTs was initially identified as a sodium-dependent inorganic phosphate transporter[14], which has delayed their discovery as vesicular glutamate transporters for years[15,16] and precluded direct electrophysiological analysis of the glutamate transport mechanism. Meanwhile mutations of dileucine-like motifs allowed its retention at the cell surface and first electrophysiological data from heterologous expression systems[17].

The biochemical characterization of their transport properties in vitro have failed to establish any substantial differences between the three isoforms[3]. Vesicular glutamate transport is driven predominantly by the membrane potential ($\Delta\Psi$) established by the vacuolar $H^+$-ATPase (V-ATPase) across the vesicular membrane[12,18,19]. One striking feature of VGLUT function is its biphasic dependency on extravesicular chloride[3,19–21]. In the absence of extravesicular $Cl^-$, glutamate uptake in isolated SVs is very low, whereas low $Cl^-$ concentrations (4 mM) strongly activate transport. At higher $Cl^-$ concentrations, gluta-mate transport is gradually inhibited. This complex biphasic dependence might result from an allosteric modulation of VGLUT by $Cl^-$ combined with a dissipation of $\Delta\Psi$ resulting from $Cl^-$ loading in SVs[3,19]. Yet, an alternative non-contradictory explanation has been put forward. The expression of VGLUT1 in heterologous systems suggested that VGLUTs might possess an intrinsic $Cl^-$ permeability[15], which recently has been directly confirmed electrophysiologically[17]. VGLUT has also been proposed to contribute to $Cl^-$ transport in isolated SVs[22]. In addition, luminal $Cl^-$ concentration influences glutamate trans-port and storage inside proteoliposomes containing VGLUT1[23] as

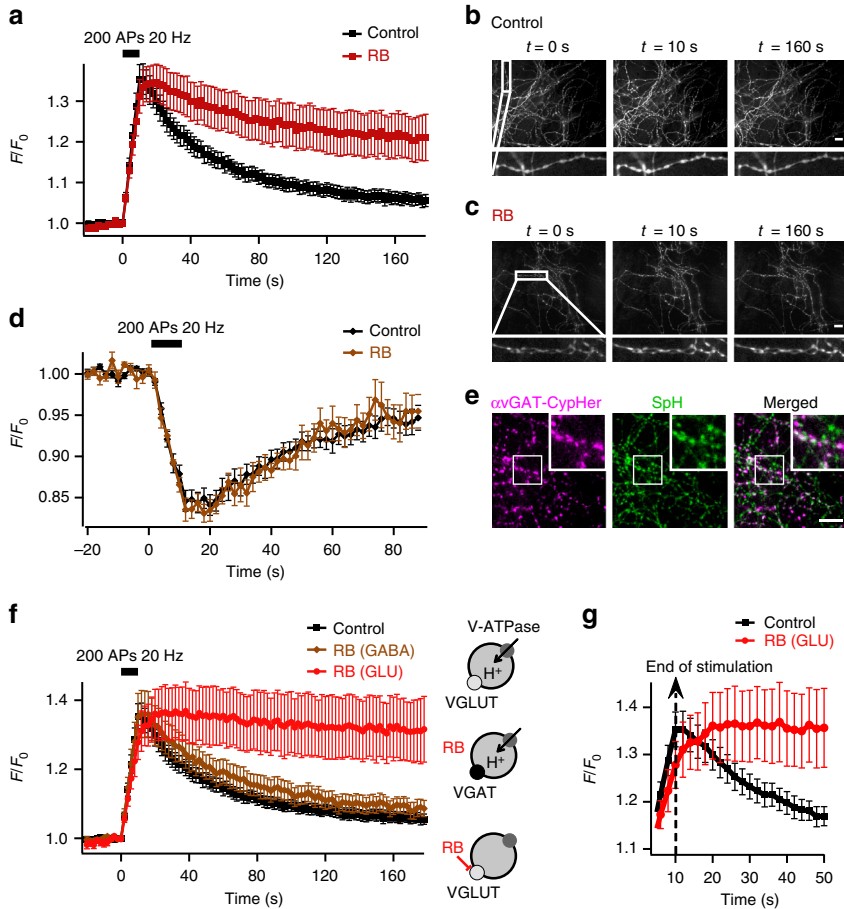

**Fig. 1** Inhibition of VGLUT with RB prevents SV re-acidification. **a** Average SpH fluorescence signals in response to 200 action potentials (APs) at 20 Hz with or without RB (100 nM, 5 min) ($n = 24$ experiments for each condition). **b**, **c** Fluorescence images of hippocampal axonal arborisations expressing SpH in control conditions **b** or in presence of RB **c** at various times after the onset of stimulation. **d** Average αvGAT-cypHer fluorescence signals in response to stimulation (200 APs 20 Hz) with ($n = 8$) or without ($n = 12$) RB. **e** Fluorescence images of SpH-transfected hippocampal neurons labelled with αvGAT-cypHer show partial colocalization with SpH at individual boutons. Scale bars in **b**–**e** represent 10 μm. **f** Effect of RB on average SpH fluorescence signals in response to 200 APs at 20 Hz in GABAergic and glutamatergic boutons ($n = 24$ for control, 11 for GABAergic boutons with RB, 13 for glutamatergic boutons with RB). **g** Enlargement of the traces in **f** at the climax of stimulation. Note the consistent post-stimulus rise in SpH fluorescence in inhibited glutamatergic boutons due to the sensitivity of inhibited SVs to the activity-dependent transient alkalinisation of the cytosol (Supplementary Fig. 4). The sketches illustrate $H^+$ movements. V-ATPase is represented by dark grey-filled circles, VGLUT by light grey-filled circles and VGAT by a black filled circle. Error bars represent s.e.m

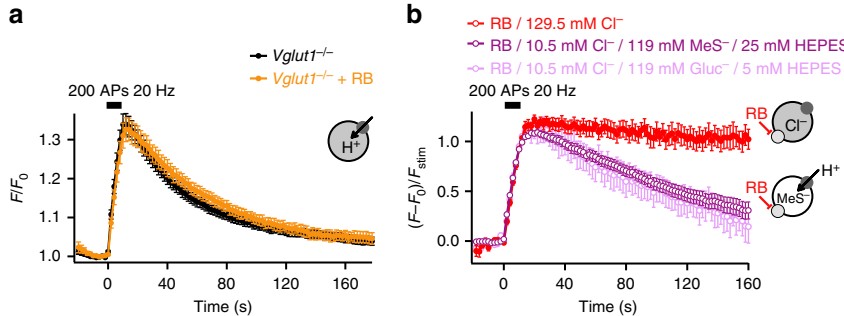

**Fig. 2** VGLUT and V-ATPase are electrically coupled. **a** SpH fluorescence time courses of glutamatergic boutons lacking VGLUT1 with ($n = 11$) or without ($n = 22$) RB. **b** SpH fluorescence time courses upon stimulation (200 APs, 20 Hz) with RB and either normal Cl⁻ concentration (129.5 mM) or a combination of 10.5 mM Cl⁻ and 119 mM MeS⁻ or Gluc⁻ ($n = 13$ and 18). Because of Gluc⁻ buffering capacity, the HEPES concentration was reduced from 25 to 5 mM when Cl⁻ was substituted with Gluc⁻ (Fig. 5). Note that glutamatergic SVs are still sensitive to cytosolic pH changes as seen by the remaining transient post stimulus increase in SpH fluorescence (Supplementary Fig. 4). The sketches illustrate H⁺ movements. The dark grey and light grey circles symbolize V-ATPase and VGLUT, respectively. Error bars represent s.e.m

well as inside isolated SVs[24], reconfirming a modulatory role of the anion suggested earlier[12]. Also protons have been suggested as allosteric modulators of VGLUT activity, since low intravesicular pH increased the transport rate, but no proton conductivity could be observed in heterologous expression systems[17] and in isolated SVs[22].

In order to study VGLUT properties in conditions that preserve the vesicular environment, we investigated the dynamics of SV acidification and neurotransmitter filling by fast live-cell imaging in situ, i.e. in cultured hippocampal neurons of wild-type and VGLUT1-deficient mice in conjunction with pharmacological interference and ion substitution of the intravesicular solution. Our results demonstrate that glutamate loading into vesicles is fuelled by ΔΨ not only produced by V-ATPase activity, but initially mostly by the channel-like chloride efflux from the vesicular lumen. Our data imply that VGLUT is a glutamate/proton exchanger non-stoichiometrically linked to a channel-like anion (Cl⁻) conductance, bestowed by the VGLUT1 protein itself. In addition, we show that glutamate transport is the rate-limiting step during SV acidification, i.e. SV acidification is the result of glutamate loading and not its prerequisite as implied by the allosteric effector model[17]. Therefore, the amount of chloride engulfed in newly endocytosed vesicles governs the re-acidification kinetics of glutamatergic SVs. Finally, our data reveal the initial transport rate of VGLUT at hippocampal glutamatergic terminals.

## Results

**Rose Bengal blocks re-acidification of newly endocytosed SVs.** To monitor the luminal pH of SV, we expressed the pH-sensitive variant of GFP, pHluorin, coupled to the luminal domain of synaptobrevin 2 (synaptopHluorin, SpH) in cultured hippocampal neurons. Electrical stimulation triggered SV fusion with the plasma membrane. Exposure of SpH to an external medium at pH 7.3 led to a fluorescence increase (Fig. 1a, b). SpH was subsequently removed from the plasma membrane by compensatory endocytosis of SVs whose re-acidification led to fluorescence decay back to baseline (Fig. 1a, b). In order to interrupt glutamate loading, we applied different known blockers of VGLUTs. We found that the most common VGLUT inhibitor Evans Blue[25] is not suitable to impact SV acidification due to its membrane impermeability (Supplementary Fig. 1). In contrast, Rose Bengal (RB), a potent noncompetitive inhibitor of VGLUTs is membrane permeant[26,27]. In presence of the scavenger histidine to protect neurons from singlet oxygen production during

illumination[28] (Supplementary Fig. 2), inhibition of VGLUT with RB induced a partial loss of SpH fluorescence recovery (Fig. 1a, c).

Next, we distinguished between SpH signals from GABAergic and glutamatergic neurons using antibodies directed against the luminal domain of the vesicular GABA transporter (VGAT) coupled to the pH-sensitive fluorescent dye CypHer5E (αVGAT-CypHer)[29] (Fig. 1e). Contrary to pHluorin, CypHer5E is quenched at neutral pH. Therefore, electrical stimulation triggered a transient decrease in αVGAT-CypHer fluorescence signals (Fig. 1d). Using dual-colour experiments with SpH and αVGAT-CypHer, we found that RB strongly and specifically affected glutamatergic but not GABAergic boutons (Fig. 1d, f), and thus not the V-ATPase at the low concentration (100 nM) used here[26]. In addition, RB did neither change exo–endocytosis rate (Supplementary Fig. 3a–c), nor affect SV recycling (Supplementary Fig. 3d, e). Thus, specific VGLUT inhibition by RB abrogated SV acidification. In addition to SV re-acidification inhibition, RB induced a transient post-stimulation SV alkalinisation (Fig. 1f, g) caused by exocytosis of V-ATPases (Supplementary Fig. 4) that transiently alkalinized cytosolic pH[30], suggesting that VGLUT inhibition renders SV sensitive to cytosolic pH changes.

**VGLUT exchanges chloride for glutamate in SVs.** We so far demonstrated that uptake of glutamate into SVs is necessary for generation and maintenance of the pH gradient, arguing that VGLUT and V-ATPase activities are functionally coupled in neurons. However, no direct molecular interaction or indirect signalling mechanism between both proteins has been described. We therefore investigated whether an electrical coupling by ions other than protons might link uptake of glutamate and vesicular acidification. Consistently, glutamatergic SV from VGLUT1-deficient mice showed normal SpH fluorescence transients upon stimulation and RB no longer blocked their re-acidification (Fig. 2a), indicating that the coupling is abolished in absence of the transporter. Interestingly, VGLUT1 itself has been shown to conduct Cl⁻[17,24]. This exchange might occur through two independent binding sites[23], thus Cl⁻ transport might be unaffected by the RB block. Such an exchange of Cl⁻ for negatively charged glutamate would be electrically neutral under control conditions (one glutamate for one Cl⁻), but inhibition of glutamate transport alone would result in a positive ΔΨ by efflux of luminal Cl⁻, sufficiently positive to block the voltage-dependent V-ATPase, thereby preventing acidification. In order to study the putative role of luminal Cl⁻, we abolished the Cl⁻ gradient

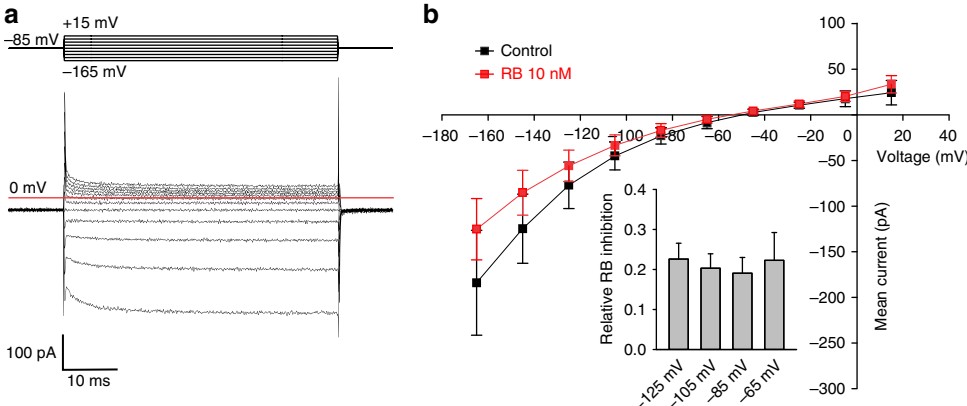

**Fig. 3** RB has only minor effects on VGLUT1 Cl⁻ currents. **a** Representative current responses from HEK cells expressing internalization-defective VGLUT1 to voltage steps from −165 mV to +15 mV in 20 mV increments from a holding potential of −85 mV. **b** Mean current voltage relationships from VGLUT1 in the absence (black square $n = 5$) and in the presence (red square $n = 5$) of 10 nM RB. Inset, relative RB inhibition at different voltages. Error bars represent s.e.m

between the vesicular lumen and the neuronal cytosol[31,32] by decreasing the extracellular Cl⁻ concentration. Substitution of Cl⁻ by an equimolar concentration of iodide affected neither SV cycling nor re-acidification (Supplementary Fig. 5). However, most Cl⁻ conductances exhibit low selectivity among inorganic anions[33]. Thus, to suppress any putative anion conductance, Cl⁻ was then substituted by either gluconate (Gluc⁻) or methanesulfonate (MeS⁻), large impermeant anions[34]. Both anions, however, display very different proton buffer capacities (pKa of 3.7 for Gluc⁻ and pKa of −1.2 for MeS⁻, respectively). Thus, the HEPES concentration was also varied for compensation. With appropriate HEPES concentrations substitution of Cl⁻ with either Gluc⁻ or MeS⁻ reversed the acidification block induced by RB (Fig. 2b), i.e. it disrupted the electrical coupling of glutamate transport to H⁺ pumping observed in the presence of RB (Figs. 1f and 2b). Since SpH response amplitudes for both GABAergic and glutamatergic boutons were reduced during Cl⁻ substitution (Supplementary Fig. 6a–d) without affecting exo–endocytosis rates (Supplementary Fig. 6f), and re-acidification in GABAergic boutons (Supplementary Fig. 6e)[35], SpH traces were normalized to their respective fluorescence values at the end of the stimulation. We conclude that in order to phenocopy the full re-acidification observed in the VGLUT1-deficient mice, block of glutamate transport by RB is not sufficient, but in addition the vesicular Cl⁻ gradient has to be abolished (Fig. 2), thus revealing a channel-like Cl⁻ conductance of VGLUT1[17] also in recycling SVs, similar to what has been observed for plasma membrane glutamate transporters of the excitatory amino acid transporter (EAAT) family[36]. Overall, these data show that Cl⁻ efflux is mostly responsible for the initial ΔΨ generation necessary for glutamate transport at neutral pH at the beginning of SV refilling. Usually this ΔΨ build-up is partially shunted by onsetting glutamate uptake, but in the presence of RB (Figs. 1f and 2b) this shunt is blocked, such that ΔΨ quickly reaches a value which stalls the voltage-dependent V-ATPase activity. In boutons from VGLUT1-deficient neurons this Cl⁻-dependent ΔΨ build-up is absent, thus re-acidification is normal (Fig. 2a). To directly demonstrate the existence of a chloride conductance in the presence of RB, we heterologously expressed mutant VGLUT1 that was optimized for surface membrane insertion in HEK293T cells[17] and performed whole-cell patch-clamp recordings at pH 5.0 in the absence as well as in the presence of RB. To assess the effect of RB on chloride efflux out of the synaptic vesicles, we established an anion gradient with a cytoplasmic [Cl⁻]ᵢₙₜ of 14 mM and an external [Cl⁻] of 145 mM (corresponding

to the high anion concentration in the vesicle after endocytosis). Under these conditions, we observed an inwardly rectifying chloride current that reversed close to the calculated anion reversal potential (Fig. 3a). RB resulted only in a reduction of anion currents (Fig. 3b).

To further confirm the pivotal role of the Cl⁻ conductance of VGLUT1 in SV refilling, we aimed to investigate the Cl⁻ concentration inside glutamatergic SV from wild-type and VGLUT1-deficient mice. First, we designed a vesicular chloride sensor based on the previously described ratiometric indicator of pH and Cl⁻ by excitation ClopHensorN[37] fused to the luminal domain of synaptobrevin 2. However synaptobrevin 2-ClopHensor was mistargeted and numerous aggregates could be observed in the neuronal cell bodies and axons. Thus, we coupled ClopHensorN to the luminal domain of synaptotagmin 1 (SytClopH) (Fig. 4a). When expressed in cultured hippocampal neurons, SytClopH exhibited a punctate distribution corresponding to its targeting to synapses (Fig. 4d). The calibration of SytClopH at various pH and [Cl⁻] values by ionophore clamping technique (Methods) indicated that the vesicular sensor conserved the properties described for the cytosolic ClopHensorN (Fig. 4b, c and Supplementary Fig. 7)[37]. A tobacco etch virus (TEV) protease cleavage site was present between synaptotagmin 1 and the ClopHensorN moieties. This site is only accessible to external enzyme if SytClopH is in the plasma membrane. Proteolytic cleavage of ClopHensorN from the plasma membrane led to a virtually pure vesicular expression of SytClopH (Fig. 4a) allowing us to directly measure the luminal pH and chloride concentration of SV. The measured $R_{pH}$ showed a similar acidic pH in the lumen of both wild-type and VGLUT1-deficient mice (Fig. 4e). The $R_{Cl}$ measured in SV from wild-type mice indicated that the average glutamatergic luminal [Cl⁻] was $14.7 \pm 2.50$ mM (Fig. 4f), similar to the [Cl⁻] within the neuronal cytosol[31,32]. The absence of a chloride gradient between the SV lumen and the cytosol at resting state is consistent with an efflux of endocytosed chloride during SV recycling (~130 mM extracellular [Cl⁻] upon vesicle formation). On the opposite, the average $R_{Cl}$ measured in SV from VGLUT1-deficient mice was significantly smaller than $R_{Cl}$ from wild-type animals, indicating that in absence of VGLUT1 the luminal [Cl⁻] is much higher and reaches values in the saturation range of SytClopH (>90% saturation, i.e. >88.3 mM) (Fig. 4f). The significant difference in luminal [Cl⁻] from wild-type and VGLUT1-deficient mice demonstrates that VGLUT1 itself is responsible for the effective removal of luminal Cl⁻ during SV recycling down to the low cytosolic concentration.

Our results imply that SVs are loaded with about 120 mM glutamate under isosmotic conditions.

**Luminal proton buffer capacity.** The most common impermeant anion used to substitute $Cl^-$ is $Gluc^-$. However, unlike $MeS^-$ used earlier, $Gluc^-$ possesses a small proton-buffering capacity leading to more complex dynamics of re-acidification which critically depend on the HEPES concentration, thus on exogenous proton buffer capacity. For the normal HEPES concentration of 25 mM used here, SpH fluorescence signals of glutamatergic boutons never returned to baseline after stimulation when $Cl^-$ was progressively substituted by $Gluc^-$ (Fig. 5a, c). This lack of acidification would be expected if the endocytosed $Cl^-$ concentration controlled the final vesicular glutamate content[23]. With reduced or completely absent $Cl^-$ efflux, glutamate uptake is restricted by osmotic pressure. Thus, less glutamate would be loaded, which in turn would shunt $\Delta\Psi$ increasingly less efficiently, thereby restraining more and more $H^+$ influx. V-ATPase pumping would cease as soon as the membrane capacitance was charged up and $\Delta\Psi$ became too positive. The degree of acidification reached at steady state then simply would depend on the total $H^+$ buffering capacity inside the SV. In line with this, decreasing the total luminal buffering capacity by reducing HEPES concentration allowed complete acidification in presence of high $Gluc^-$ concentrations (Fig. 5b). The $H^+$ buffering capacity of 119 mM $Gluc^-$ ($pKa$ of 3.7) at pH 5.6 corresponds roughly to the buffering capacity of a few mM HEPES ($pKa$ of 7.55). Thus, a reduction to 20 mM HEPES led already to re-acidification to pH~6.3, and a reduction to 5 mM HEPES to full re-acidification (Fig. 5e). Accordingly, substitution by $MeS^-$ which has no $H^+$ buffering capacity ($pKa$ of −1.2) led to full re-acidification even in presence of 25 mM HEPES (Fig. 5d). These data show that $Cl^-$ efflux from the vesicular lumen controls the reacidification kinetics, which are considered to be a proxy for glutamate loading. Thus, $Cl^-$ efflux appears to control the amount of glutamate loaded into SVs in line with a previous study[23], and thereby also the final $H^+$ concentration. As a consequence, the degree of acidification depends on the intravesicular $H^+$ buffering capacity.

**VGLUT is a glutamate/$H^+$ exchanger.** To further characterize the role of $Cl^-$ and $H^+$ flux coupling for glutamate loading, we next analysed the acidification kinetics using the 'rapid acid quench' strategy[38]. In this paradigm, the membrane of hippocampal neurons expressing SpH was quenched by acidic buffer shortly after a train of electrical stimulation (Fig. 6). Newly endocytosed SpH in not-yet acidified SV was protected from the surface quench while subsequent compensatory endocytosis was masked by acid. Therefore, the fluorescence decay during the quench, absent if not stimulated, directly reflects re-acidification kinetics (Fig. 6b, c). The time course of re-acidification is well approximated by a mono-exponential with a time constant of 4.9 ± 0.3 s. Notably, neither histidine nor HEPES at the concentrations used (up to 25 mM) influenced significantly the time course of re-acidification (compare with controls without histidine in Fig. 7 and Supplementary Fig. 8). However, with 40 mM HEPES the SV re-acidification is somewhat slowed down (Supplementary Fig. 8), suggesting that the endogenous buffer capacity roughly equals that of a few 10 mM HEPES. This is in line with a recent theoretical estimate of the luminal buffer capacity provided by the ~560 glutamate and ~150 histidine residues in the luminal protein matrix, yielding a buffer capacity of 10–15 mM per pH unit[39]. The impact of high mM of exogenous buffer on the re-acidification rate is confirmed by the use of TRIS as an alternate proton buffer (Supplementary Fig. 8). Consistent with Fig. 1f, in presence of RB, the fluorescence during acid quench did not

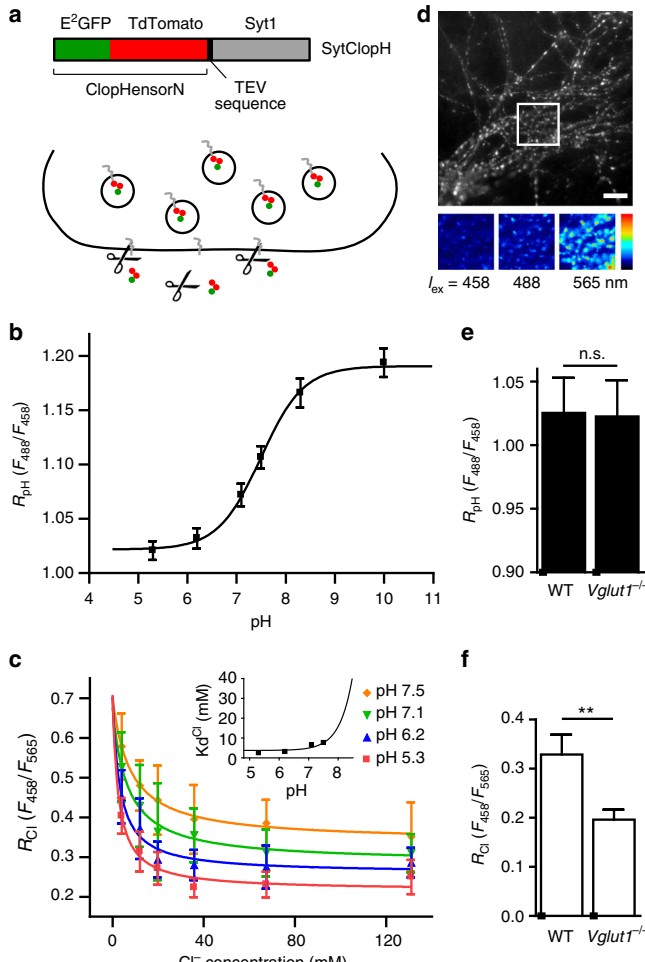

**Fig. 4** Absence of VGLUT1 prevents the efflux of luminal $Cl^-$. **a** The synaptotagmin 1-ClopHensorN (SytClopH) construct is composed of the previously described ClopHensorN ($E^2$GFP-TdTomato fusion)[37] fused to the cleavage site for recombinant TEV protease and synaptotagmin 1 (Syt1) (top). When neurons-overexpressing SytClopH were exposed to 60 U $ml^{-1}$ TEV protease (scissors) at room temperature, ClopHensorN was cleaved from molecules residing in the plasma membrane while vesicular SytClopH was inaccessible to the protease (bottom). **b** SytClopH calibration curve relating pH to the ratio of fluorescence emission when $E^2$GFP was excited at 488 nm over the emission when excited at 458 nm ($R_{pH}$). $pKa$ was found to be 7.49 with a fitted 95% confidence interval between 7.23 and 7.76. Note that variation of $Cl^-$ concentration within the physiological range had no effect on RpH (Supplementary Fig. 7). **c** SytClopH calibration curve relating $Cl^-$ concentration to the ratio of fluorescence emission when $E^2$GFP was excited at 458 nm over the emission when TdTomato was excited at 565 nm ($R_{Cl}$). Change in pH influences the affinity of SytClopH for $Cl^-$. **d** Fluorescence image of hippocampal axonal arborisations of wild-type neurons expressing SytClopH collected following excitation at 565 nm. The insets (bottom) show a higher magnification with excitation at 458, 488 and 565 nm respectively. Scale bar represents 10 μm. **e, f** Histogram of the measured $R_{pH}$ (**e**) and $R_{Cl}$ (**f**) in hippocampal neurons from wild-type (WT, $n = 29$) and VGLUT1 knock-out ($Vglut1^{-/-}$, $n = 27$) mice. n.s. not significant $P = 0.9447$; **$P = 0.0060$ analysed with two-tailed unpaired $t$-test. Error bars represent s.e.m

decay and had a higher intensity compared to background, revealing an alkaline, not re-acidified pool of endocytosed SVs (Fig. 6c).

As might be expected from the slow decay kinetics observed for $Gluc^-$ and $MeS^-$ substitution (Fig. 5), SV acidification determined with the acid quench was significantly slowed by $Cl^-$ removal

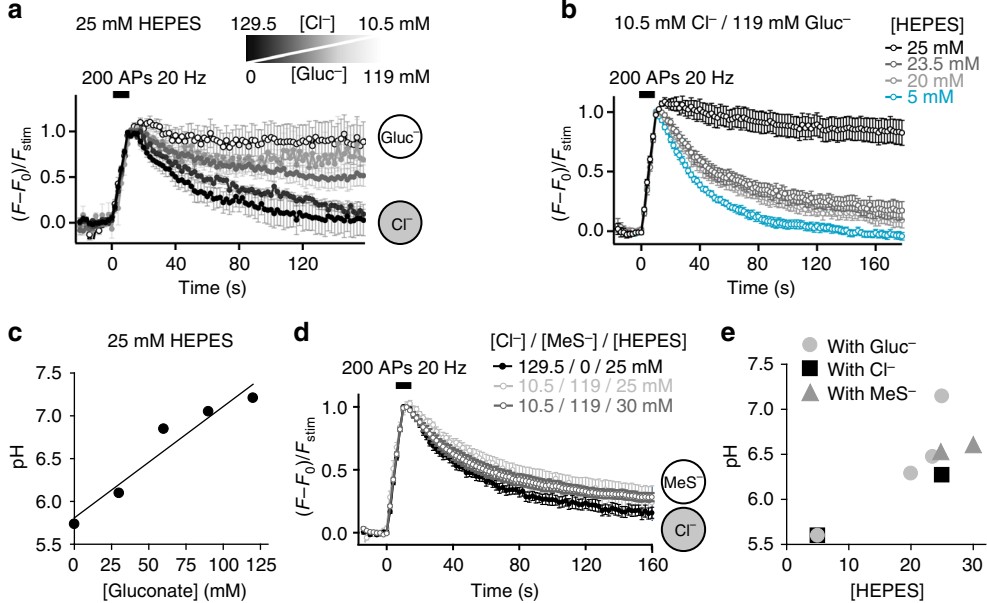

**Fig. 5** Dependence of SV acidification on luminal buffering capacity. **a** Average SpH transients in response to stimulation (200 APs, 20 Hz) in neurons exposed to different combinations of extracellular chloride/gluconate concentrations ([Cl⁻]/[Gluc⁻]) (n = 3–10). Cl⁻ concentration is decreased from 129.5 mM (black) to 100–70–40–10.5 mM while being substituted by Gluc⁻ to maintain a final anion concentration of 129.5 mM. The sketches show the main anion present in SV. **b** Average SpH fluorescence responses to 200 APs at 20 Hz in presence of 10.5 mM Cl⁻, 119 mM Gluc⁻ and different HEPES concentrations as indicated (n = 6–14). **c** Luminal pH of SVs endocytosed with various Gluc⁻ concentrations, estimated from the plateaus reached in **a**. **d** Average SpH responses to stimulation (200 APs, 20 Hz) in neurons exposed to different HEPES concentrations and different combinations of extracellular chloride/methanesulfonate concentrations ([Cl⁻]/[MeS⁻]) (n = 7–16). The sketches show the main anion present in SV. **e** Luminal pH of SVs endocytosed with various HEPES and ion concentrations, estimated from the plateaus reached in **a**, **b**, **d**. Traces in **a**, **b**, **d** represent responses of glutamatergic boutons only. Error bars represent s.e.m

(Fig. 7). Most notably, the acidification kinetics were equally slow, irrespective of the substitute (Gluc⁻ or MeS⁻) or HEPES concentration used, i.e. irrespective of the H⁺ buffer capacity and the steady state pH reached. This finding implies that H⁺ influx through the V-ATPase is mostly shunted away by a H⁺ efflux of almost equal size, and that the acidification kinetics merely reflect the time for reaching a steady state.

In order to unmask this massive H⁺ antiport associated with VGLUT activity, we investigated the re-acidification kinetics under conditions where VGLUT and V-ATPase activities are uncoupled: VGLUT1 knockout or RB plus Cl⁻ substitution. In the absence of Cl⁻ efflux and glutamate influx, acidification was accelerated by at least an order of magnitude (Fig. 8). A deficit in endocytosis and/or acidification in VGLUT1 knockout synapses as alternative explanation could be ruled out. First, the SpH fluorescence fully recovered to baseline (Fig. 2a). Second, the plateaus during the acid quench with and without stimulation, i.e. with and without exo-/endocytosis of synaptic vesicles, were identical indicating that all vesicles have been re-acidified to steady state (Fig. 8c). Interestingly, under these uncoupling conditions the H⁺ flux through the V-ATPase can directly build up the pH gradient and the degree and kinetics of acidification only depend on the luminal H⁺ buffering capacity and the electrical membrane capacity. Indeed, lowering the concentration of HEPES from 25 mM to 5 mM now led to an even faster acidification, not easily distinguishable from the solution exchange time course during acid quench (Fig. 8b inset and 8e).

## Discussion
Using hippocampal neurons in culture, Cl⁻- and pH-sensitive fluorescent sensors, we found that in SVs VGLUT1 functions as a glutamate/H⁺ exchanger associated with a stoichiometrically uncoupled Cl⁻ conductance. As a consequence, Cl⁻ taken up

during endocytosis fuels ΔΨ-driven glutamate uptake and consequently controls SV acidification velocity (Fig. 9). In the beginning of SV recycling, there is no proton gradient and considering the SV volume, a pH of 7.3 corresponds to only 0.5‰ of a single free proton. VGLUT1 is a voltage- and H⁺-dependent antiporter, i.e. in order to pump glutamate it needs either a positive voltage or (at lower voltage) a proton gradient. At first, it is thus the ΔΨCl that drives VGLUT1, and ~10 transporters per SV pump as much H⁺ out as the ~1.5 V-ATPases per SV pump in. The high voltage (ΔΨCl) also impedes the V-ATPase pumping speed, thus the V-ATPase just keeps pace with the VGLUTs. As the Cl⁻ gradient decreases VGLUTs slow down, but this is now compensated by the accelerating V-ATPase activity which leads to a VGLUT1-stimulating pH decrease. Still, at this point, essentially every H⁺ pumped in by the V-ATPase is transported out by VGLUT1 (even pH 5.5 corresponds to only 5% of one free proton) in exchange to glutamate. Thus, as ΔΨCl vanishes, ΔΨH builds up and takes over the energizing glutamate transport, allowing the V-ATPase to acidify the lumen of the vesicle. Coupling glutamate influx with Cl⁻ efflux as counter anion ensures osmotic and electrical neutrality of the vesicular filling process, and in the first place enables concentrating glutamate up to an isosmotic, i.e. maximum concentration of up to 1800 molecules in the SV[40].

Our results show that the electrical coupling between acidification and glutamate transport reflects a glutamate/H⁺ exchange mechanism for VGLUT activity (Fig. 9). The massive H⁺ antiport associated with VGLUT activity enabled us to infer the glutamate transport rates and properties from the SV acidification measurement. We thus demonstrated that vesicular acidification rates scale with luminal chloride concentrations, in accordance with the fact that high chloride concentration in VGLUT1-containing proteoliposomes enhanced glutamate transport

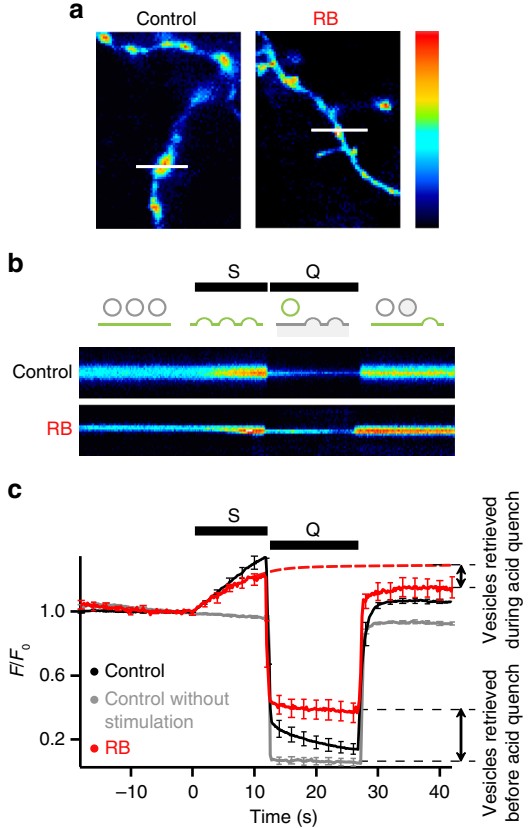

**Fig. 6** Time course of SV acidification. **a** SpH fluorescence of axons and presynaptic terminals after stimulation (12 s, 20 Hz) in control conditions or after 5 min incubation with 100 nM RB. Lines represent the position of the scan lines in **b**. **b** Line-scan time series of single boutons. S: field electrical stimulation, Q: rapid surface quenching by application of impermeant acid (pH 5.25) immediately after end of stimulation. Pseudo-colour scale bars **a**, **b** are linear, with black indicating zero fluorescence. Schematic shows the fluorescence of three SVs recaptured at different rates: before, during and after acid quench (from left to right) in control conditions. **c** Average SpH responses without ($n=10$) or with stimulation in absence ($n=11$) or presence ($n=6$) of RB. Red dashed line represents the average SpH response in presence of RB, without acid quench. Time scale is the same in **b** and **c**. Error bars represent s.e.m. They are shown every tenth point (2 s) for better visibility

rates[23]. Accordingly, in control conditions the acidification time constant does not depend on HEPES concentration, which was previously observed but interpreted as an exclusion of HEPES from SV undergoing endocytosis or the inability of buffering $H^+$ in the microenvironment of SV[38]. A few mM of HEPES or TRIS can already increase sufficiently the buffering capacity of SVs and induce slower acidification. Although TRIS has a higher p$K$a than HEPES (8.3 and 7.55 respectively) and thus a higher buffering potency, HEPES affected the reacidification kinetics more at 5 mM, in line with a previous observation[38]. It is unclear what property, their charge or size, is responsible for this difference.

The glutamate/$H^+$ exchange facilitates the transport of glutamate against its gradient and fuels this transport when the $Cl^-$ gradient vanishes. Without this strictly coupled proton antiport SVs could only be filled half, i.e. up to the point where $Cl^-$ and glutamate gradients are equal and opposite in direction. Accordingly, V-ATPase inhibition decreases glutamatergic evoked and miniature excitatory postsynaptic currents in cultured hippocampal neurons although the chloride conductance produces most of the $\Delta\Psi$ required for glutamate loading[41,42]. The

glutamate refilling process therefore can be separated into two phases. Initially the $Cl^-$ gradient is large and thus $\Delta\Psi$ is high, close to the $Cl^-$ reversal potential. Under this high-$\Delta\Psi$ condition the voltage-dependent V-ATPase activity is low, while the VGLUT activity is high and fuelled mostly by the $Cl^-$ conductance. Most protons imported by the V-ATPase are exported by the VGLUT1 transporters. When the $Cl^-$ gradient, however, decreases towards half its initial value, $\Delta\Psi$ would vanish. But this decreasing membrane potential now disinhibits the V-ATPase so that further glutamate can be transported in this second phase. Although the membrane potential further drops VGLUT activity is now maintained by progressively lowering pH, i.e. V-ATPase activity at low membrane potential now exceeds that of VGLUT1. In this way the $Cl^-$ gradient can be further decreased down to close to zero. Eventually at pH 5.5 the V-ATPase activity is blocked by the almost 100-fold $H^+$ gradient, and the refilling process comes to an end.

In line with this notion SV acidification was strongly accelerated in VGLUT1-deficient mice. Previous experiments have reported that ATP in presence of glutamate induced a strongly reduced acridine orange fluorescence quenching in SVs isolated from VGLUT1-deficient compared to wild-type mice[23]. Acridine orange fluorescence quenching is often used as a proxy of SV acidification. Thus the authors concluded that the absence of VGLUT1 impaired acidification of isolated SVs, in apparent conflict with our results. However, due to the likely loss of all ion gradients during the isolation procedure, we expect a net influx of glutamate without any coupled anion efflux in wild-type isolated SVs. This accumulating negative charge could be dissipated by massive $H^+$ loading, leading to an even lower luminal pH compared to SVs in intact cells and isolated from VGLUT1-deficient neurons. Notably the acridine orange fluorescence quenching assay does not report absolute pH values, and acidification in isolated SVs requires minutes instead of seconds, supporting this notion. Finally, the discrepancy between living cells and isolated SVs might hint at another yet to be identified ion transport mechanism in SVs which can dissipate to some degree $\Delta\Psi$ generated by the V-ATPase in SVs in intact cells, but not in isolated SVs where the ionic composition is different. This may explain why we observe rapid acidification in the absence of VGLUT, while positive charge accumulation ($\Delta\Psi$) in isolated SVs prevents further acidification.

The role of cations cannot be easily studied in living neurons, as their concentrations cannot be changed without massively affecting excitability and thus exo–endocytosis, or the ATP homeostasis when using ionophores. The role of $K^+$ and $Na^+$, however, has been investigated on isolated vesicles. For instance, the modulatory effect of $K^+$ on glutamate uptake in isolated vesicles or proteoliposomes is abolished when $Cl^-$ is present in the lumen[24]. So even if we cannot exclude a role for $K^+$ at the end of the loading process when the $Cl^-$ gradient is abolished, we do not expect a major role of this cation under physiological conditions. Additionally, no effect of $Na^+$ on glutamate uptake could be detected in the reconstituted system[24]. Finally, VGLUT2 was shown to contain two independent machineries for the $Cl^-$-dependent uptake of glutamate and the $Na^+$-dependent uptake of phosphate[20]. Therefore, cations like $K^+$ and $Na^+$ are likely to have only minor effects if any, on VGLUT activity and glutamate transport.

When VGLUT and V-ATPase activities are uncoupled, measured acidification time constants scaled roughly proportionally to the respective exogenous buffering capacities (Fig. 8), corroborating the notion that the endogenous SV buffering system is rather small and of comparable size. Thus, with a reasonable estimate of the buffering capacity, i.e. the number of buffering molecules for the different conditions, we can quite well infer the

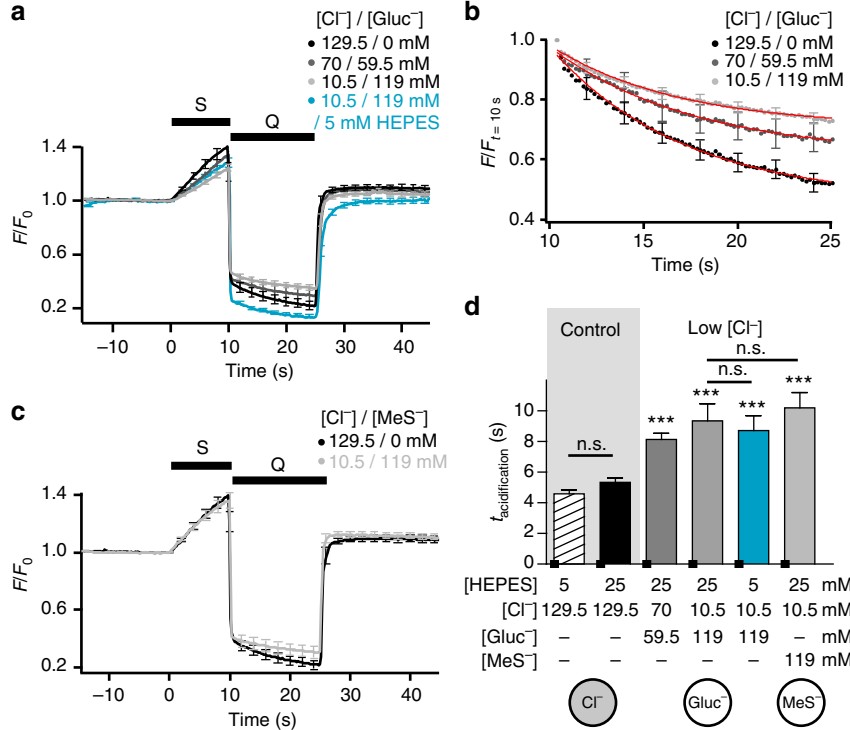

**Fig. 7** Vesicular chloride controls the rate of acidification. **a–c** Rapid acid quench (Q) of surface SpH fluorescence following stimulation (S, 200 APs at 20 Hz) in presence of different $[Cl^-]/[Gluc^-]$ (**a**, **b**) or $[Cl^-]/[MeS^-]$ (**c**) with 25 mM HEPES unless otherwise stated ($n = 8–17$). **b** The fluorescence from SpH shown in **a** was normalized to the first data point measured during post-stimulus acid quench. Red curves: single exponential fits of the SpH fluorescence during the quench. **d** Bar graph of acidification time constants from experiments in **a** and **c**. 25 mM HEPES: $P = 0.0676$ compared with 5 mM HEPES; 70 mM $Cl^-$/59.5 mM $Gluc^-$: $P < 0.0001$ compared with 25 mM HEPES; 10.5 mM $Cl^-$/119 mM $Gluc^-$: $P = 0.0001$ compared with 25 mM HEPES; 5 mM HEPES/10.5 mM $Cl^-$/119 mM $Gluc^-$: $P < 0.0001$ compared with 5 mM HEPES, $P = 0.6936$ compared with 25 mM HEPES/10.5 mM $Cl^-$/119 mM $Gluc^-$; 10.5 mM $Cl^-$/119 mM $MeS^-$: $P < 0.0001$ compared with 25 mM HEPES, $P = 0.5703$ compared with 10.5 mM $Cl^-$/119 mM $Gluc^-$; analysed with two-tailed unpaired $t$-test. Sketches below show the main anion present in SV. Traces represent responses of glutamatergic boutons only. n.s. non significant; ***significant changes. Error bars represent s.e.m. They are shown every tenth point (2 s) for better visibility

initial $H^+$ pumping rates from the acidification rate constants. Upon recycling, the pH of SV decreases from 7.3 to 5.5. Within $\tau$ seconds the SpH fluorescence intensity drops to $1/e$ of its initial value, which roughly corresponds to a drop in pH to about 6.4 according to the measured pH-dependence of pHluorin[43]. SVs have an internal volume of $20 \times 10^{-21}$ l[40]. Using the Henderson–Hasselbalch equation (pH = pKa + log([A$^-$]/[HA])) and taking into account 25 mM HEPES (pKa=7.55) and 10 mM histidine (pKa=6) as the only exogenous effective buffers (pHluorin itself with one to two copies per SV only[44,45] and MeS$^-$ are negligible), and 13 mM histidine and 47 mM glutamate (pKa=4) residues in the luminal protein matrix as relevant endogenous buffers[39] we calculated that ~150 $H^+$ are bound to free buffer molecules during this time, while for the small SV volume the number of free protons is well below one molecule. With a time constant $\tau$ of $0.5 \pm 0.1$ s (Fig. 8b, e) and ~1.5 copies of V-ATPase per SV[40] this suggests an initial $H^+$ pumping rate of ~200 s$^{-1}$ per V-ATPase, i.e. ~34 rotations per second of the $V_0$ subunits, well in the range of rates measured in vitro[46]. In another recent study the authors aimed at determining the vesicular proton buffer capacity by titration with different ammonium concentrations[47]. Using a simple buffer equation, assuming only mass balance for $NH_3$, $NH_4^+$ and protons inside and outside the vesicle, they arrived at a significantly higher estimate for the endogenous buffer capacity of 57 mM $NH_4^+$ per pH unit. However, this simple model neglects all other possible ion fluxes induced during titration, like by vATPase activity triggered by pH neutralization, and thus should lead to an overestimation of the buffer capacity.

Under control conditions, a final vesicular glutamate concentration of 120 mM is exchanged for 120 mM $Cl^-$ according to our model, thus about 890 glutamate molecules would be loaded into a SV within $\tau$~5 s. With about 10 VGLUTs per vesicle[40], this yields an initial transport rate of ~20 molecules per second per transporter in intact neurons. Assuming the same $H^+$ transfer rate under control conditions, the measured acidification time constant of ~5 s would imply an influx of ~1500 $H^+$ during this time. Taking into account the endogenous (histidine and glutamate) and exogenous (HEPES) buffers as well as the quantity of glutamate loaded, we found that ~100 $H^+$ are buffered in the lumen within $\tau$ seconds, also indicating that ~1400 $H^+$ are free. Since the electrical membrane capacitance of about 50 aF per SV[48,49] means that as few as 30 $H^+$ generate already a membrane potential of 100 mV, most of this charge obviously must be shunted away by a $H^+$ efflux of roughly equal size, i.e. 1400 $H^+$. Considering that $Cl^-$ efflux initially impedes the V-ATPase pumping speed during glutamate loading and that the rough calculation of the total luminal buffer capacity is presumably underestimated[22,47], our data are compatible with a 1:1 to 1:2 stoichiometrically coupled glutamate/$H^+$ exchange through VGLUT.

Based on its reconstitution in liposomes VGLUT1 has been proposed to exchange luminal chloride for glutamate through two independent binding sites, thereby stimulating glutamate transport[23]. In line, in a heterologous expression system a stoichiometrically uncoupled $Cl^-$ conductance has been revealed electrophysiologically[17]. Accordingly, we found that $\Delta\Psi$ generated by luminal chloride efflux was not blocked by RB

(Figs. 1 and 3). Thus, the chloride transport must depend upon conformational changes distinct from those required for glutamate uptake, similar to the substrate-activated anion channel function found in membrane glutamate transporters of the EAAT family[36]. However, the finding of this anion conductance does not exclude a role for cytosolic chloride as an allosteric effector of VGLUT activity[12,17,24].

In addition to VGLUT ClC-3 has been suggested to transport chloride in SVs[50]. ClC-3 is a $Cl^-/H^+$ exchanger which has been proposed to provide a shunt for proton pump currents in SVs, thus allowing acidification[50]. As such, ClC-3 could modulate VGLUT activity. The presence of ClC-3 in glutamatergic SVs is still a matter of debate[23,40,50]. Nevertheless, by loading $Cl^-$ in SVs, ClC-3 would shunt the $Cl^-$ conductance of VGLUT, eventually dissipate $\Delta\Psi$ and subsequently stop VGLUT activity.

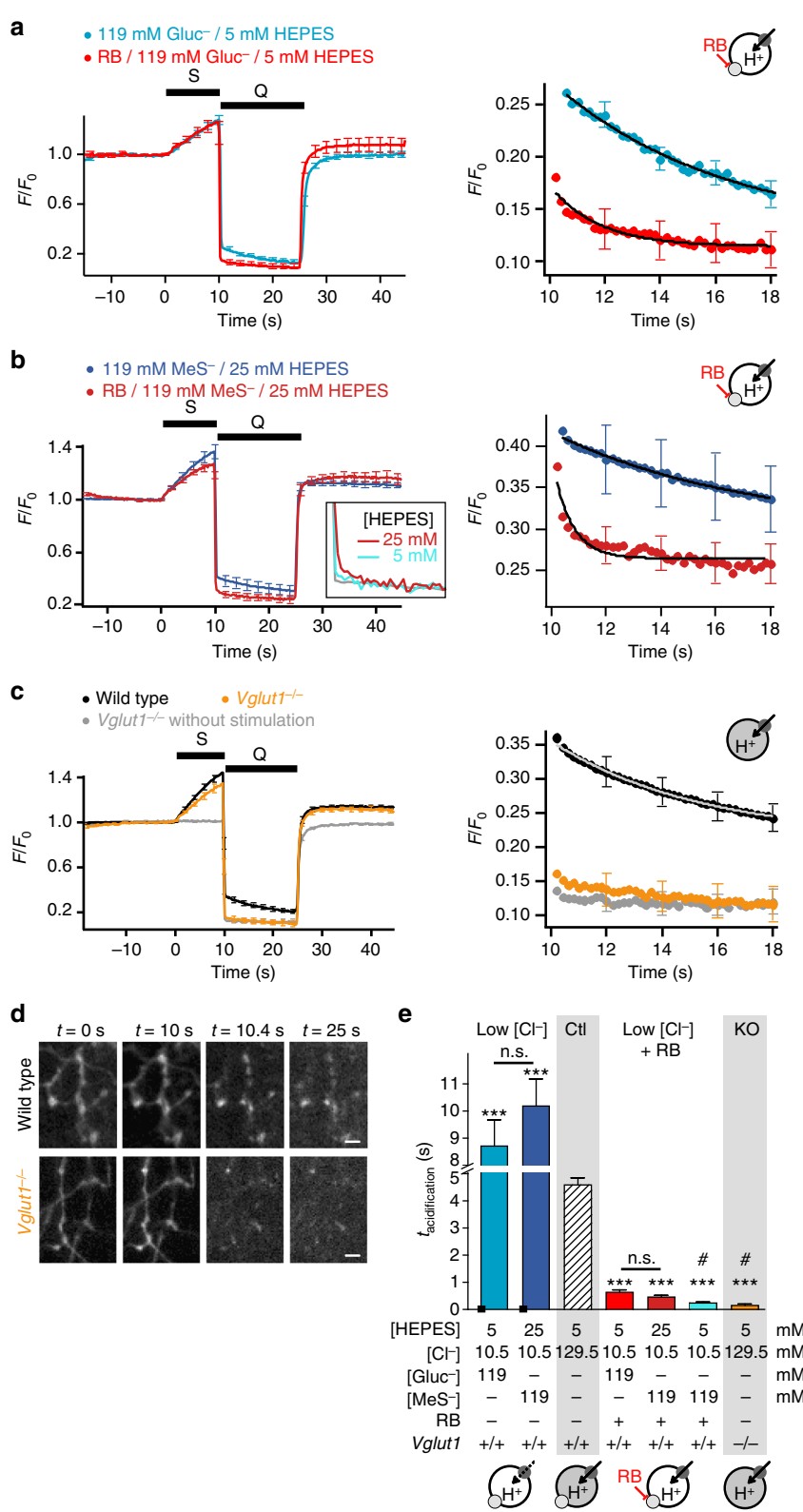

Accordingly, ClC-3$^{-/-}$ mice show a slight increase in the amplitude and frequency of miniature excitatory postsynaptic currents as well as in the amplitude of evoked excitatory postsynaptic currents, suggesting an increase in vesicular glutamate filling consistent with the observed excitotoxicity and loss of the hippocampus formation[51,52]. Therefore, ClC-3 seems to fine tune and even rather restrain the glutamate loading into SVs.

Our kinetic data imply that one single VGLUT1 copy with a transport rate of 20 s$^{-1}$ in conjunction with a single V-ATPase is sufficient to fully load a SV within 1 min. Indeed, it has been shown in hypomorph VGLUT Drosophila mutants that a single transporter molecule suffices[53]. Other studies, however, showed that the neurotransmitter content is to some extent controlled by the transporter copy number[5,6,27]. This has been interpreted as a result of an equilibrium between glutamate uptake and leakage. Our finding of a channel-like Cl$^-$ conductance in VGLUT1 offers an alternative explanation: the final filling level could be controlled by modulating the Cl$^-$ fluxes, i.e. the amount of Cl$^-$ flowing out in exchange for glutamate. The mechanism of vesicular glutamate transport is particularly interesting since V-ATPase was proposed to act as an acidification sensor in order to regulate exocytosis in hippocampal neurons and neurosecretory cells[54]. Because glutamate loading controls the rate of luminal acidification, V-ATPase not only senses the vesicular pH but it also indirectly senses whether the filling process is still ongoing or completed. Therefore, VGLUT signals to the exocytotic machinery whether a vesicle is ready for release using V-ATPase as an intrinsic link between vesicular fill state and fusion potency. Accordingly, the intravesicular glutamate fill state content was suggested to regulate SV release probability at hippocampal synapses in culture[55].

Ever since the quantal hypothesis of transmitter release was proposed there has been lively debate about whether modulation of VGLUT activity might drive changes in synaptic strength. The glutamate/Cl$^-$ exchange mode identified here represents an important means to control quantal content (Fig. 9d). The physiological as well as pathological conditions which lead to variations in extracellular and intravesicular Cl$^-$ remain to be determined in order to ascertain that these Cl$^-$ changes impact glutamatergic quantal size and consequently synaptic plasticity in vivo.

## Methods

**DNA construct.** The original ClopHensorN construct[37] was kindly provided by Colin Akerman (Addgene plasmid #50758). A site-directed mutagenesis of ClopHensorN was performed to introduce a HindIII restriction site at the beginning of the ORF using the primers: SDM_ClopH_HindIII_fwd and SDM_ClopH_HindIII_rev (Supplementary Table 1). Then, a PCR was performed to add a XmaI restriction site at the end of the ClopHensorN-HindIII sequence using the primers: ClopH_HindIII_fwd and ClopH_XmaI_rev (Supplementary Table 1). The synaptotagmin 1-ClopHensorN (SytClopH) construct was finally made by replacing the pHluorin moiety from a synaptotagmin 1-TEV-pHluorin construct[56] with the

cDNA encoding the ClopHensorN (Addgene). The new construct maintained the cleavage site for recombinant TEV protease (rTEV) flanked by spacer arms (amino acid sequence DYDIPTTLENLYFQGELKTVDAD). The construct was verified by dideoxynucleotide sequencing.

**Cell culture.** All animals were treated in accordance with the regulations and guidelines of the State of North Rhine-Westphalia. Dissociated cultures of hippocampal neurons were prepared from the CA3/CA1 region of 1- to 3-day-old CD1 wild-type or VGLUT1-deficient (purchased from the Mutant Mouse Regional Resource Center University of North Carolina, MMRRC UNC, supported by the NIH) mice. Cells were plated on Matrigel-coated coverslips in plating medium (MEM supplemented with 25 mM glucose, 2.5 mM NaHCO$_3$, 100 µg ml$^{-1}$ transferrin, 10% FCS, 2 mM glutamine and 25 µg ml$^{-1}$ insulin), and maintained at 37 °C and 5% CO$_2$ in growth medium (Neurobasal-A medium supplemented with B27 (Gibco) and 2 mM glutamine)[57]. Transfection of supereclipic SpH and SytClopH was performed at 4 days in vitro (DIV) by a modified calcium phosphate transfection procedure[56]. Experiments were carried out at 15–25 DIV.

**Recycling SV staining.** In addition to SpH expression, SVs were labelled with FM1-43 (Invitrogen) and αVGAT-CypHer (Synaptic System). Neurons were exposed to 5 µM FM1-43 during and for 5 min after stimulation (900 APs, 20 Hz), and then washed thoroughly before imaging. To label SVs from GABAergic neurons, cells were incubated with αVGAT-CypHer in a 37 °C incubator for 3–4 h in a carbonate buffer containing 105 mM NaCl, 20 mM KCl, 2.5 mM CaCl$_2$, 1 mM MgCl$_2$, 10 mM glucose, 18 mM NaHCO$_3$. Cells were then washed three times before imaging.

**Epifluorescence microscopy of living neurons.** All experiments, unless otherwise stated, were carried out in 119 mM NaCl, 2.5 mM KCl, 2 mM CaCl$_2$, 2 mM MgCl$_2$, 30 mM glucose, 25 mM HEPES pH 7.3. Neurons were stimulated by electric field stimulation (platinum electrodes, 10 mm spacing, 1 ms pulses of 50 mA and alternating polarity at 20 Hz) applied by constant current stimulus isolator (WPI A 385, World Precision Instruments) in the presence of 10 µM 6-cyano-7-nitroquinoxaline-2,3-dione (CNQX) and 50 µM D,L-2-amino-5-phosphonovaleric acid (AP5) to prevent recurrent activity. Fast solution exchanges were achieved through a three-barrel glass tubing perfusion system controlled by a piezo-controlled stepper device (SF778, Warner Instruments). In 'rapid acid quench' experiments the system was optimized for a fast and complete acid quench by adjusting the position of the glass pipette. Ammonium chloride solution (pH 7.3) was prepared by replacing 50 mM NaCl with NH$_4$Cl, while all other components remained unchanged. Acidic solution (pH 5.25) was prepared by substituting HEPES with 45 mM 2-(N-morpholino)ethane sulphonic acid (MES). Chloride concentration was varied using different proportions of NaCl and sodium gluconate, sodium iodide or sodium methanesulfonate as stated while maintaining Na$^+$ concentration fixed at 119 mM. Chloride substitution is performed shortly before beginning of the measurement to minimize the effect on cytosolic anions. Rose Bengal (100 nM, 5 min), Folimycin (65 nM, 1 min) and Tetanus Neurotoxin (TeNT, 10 nM, 16–18 h) were added where indicated. Histidine (10 mM) was added to buffers in all experiments were RB was involved. In experiment where Ca$^{2+}$ is omitted, Ca$^{2+}$ was replaced by an equimolar concentration of Mg$^{2+}$ and 2 mM EGTA was added.

Experiments were performed at room temperature on an inverted microscope (Axiovert S100TV, Zeiss) equipped with a ×63, 1.2 NA water-immersion objective. Images were acquired with a cooled CCD camera (Sensicam QE, PCO) controlled by TILLvisION software (TILL Photonics) in 2 × 2 binning mode resulting in 688 × 520 pixels. CypHer5E was excited at 640 nm and SpH or FM1-43 at 480 nm with a computer controlled monochromator (Polychrom II, Till Photonics). Fluorescence was detected after passing a FITC/Cy5 dual-band filter set (AHF Analysentechnik AG). Time lapse images were acquired at 0.5 or 5 Hz with integration times from 50 to 300 ms. For dual-colour recordings alternating images in green and red channels were acquired.

**Fig. 8** VGLUT activity slows SV acidification. **a–c** Rapid acid quench (Q) of surface SpH fluorescence following stimulation (S, 200 APs at 20 Hz) after Cl$^-$ substitution with Gluc$^-$ (**a**) or MeS$^-$ (**b**) with or without RB (100 nM, 5 min) ($n = 8$–17) and in wild-type or VGLUT1-deficient boutons (**c**) ($n = 41$ for wild type, 27 for Vglut1$^{-/-}$ and 13 for Vglut1$^{-/-}$ without stimulation). Inset in **b**: higher magnification (×3) of representative SpH traces in presence of RB, 10.5 mM Cl$^-$ and 119 mM MeS$^-$ with normal (25 mM, red, $n = 8$) or low (5 mM, blue, $n = 14$) HEPES concentration, compared to the fluorescence intensity change due to acidic buffer only (grey). Right panels: higher magnification of the acidification kinetics during the acid quench, fitted with single exponentials. Traces in **a–c** represent responses of glutamatergic boutons only. **d** Fluorescent images of SpH-transfected neurons from panel **c** at various time after the onset of stimulation. Scale bars represent 2 µm. **e** Bar graph of acidification time constants in different conditions presented in **a–c**. ***$P <$ 0.0001 compared with the respective HEPES concentration with or without chloride substitution; 25 mM HEPES/10.5 mM Cl$^-$/119 mM MeS$^-$: $P = 0.5703$ compared with 5 mM HEPES/10.5 mM Cl$^-$/119 mM Gluc$^-$; 25 mM HEPES/10.5 mM Cl$^-$/119 mM MeS-/RB: $P = 0.1825$ compared with 5 mM HEPES/10.5 mM Cl$^-$/119 mM Gluc$^-$/RB; analysed with two-tailed unpaired t-test. Sketches in **a–c** and **e** illustrate H$^+$ movements. Dark grey and light grey circles symbolize V-ATPase and VGLUT, respectively. Cl$^-$-filled SVs are represented by grey-filled circles and Gluc$^-$ or Mes$^-$-filled SVs by open circles. n.s. non significant; *** significant changes; # the time constants for 5 mM HEPES, MeS$^-$, RB (inset in **b**), and for Vglut1$^{-/-}$ (**c**) are overestimated due to traces with an acidification too fast to be separated from the acid quench and fitted (3/14 and 9/27 traces, respectively). Error bars represent s.e.m. They are shown every tenth point (2 s) for better visibility

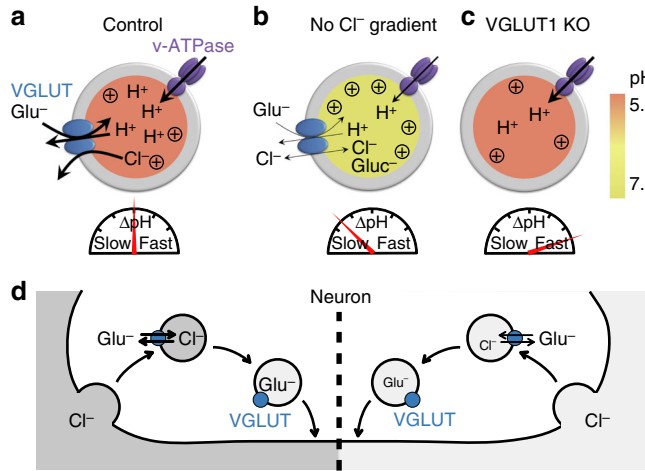

**Fig. 9** Model for VGLUT transport mechanism and SV re-acidification. Luminal pH is indicated by the green-red scale. Re-acidification rates are represented by a speedometer. Strength of $\Delta\Psi$ is symbolized by (+). **a** In presence of high luminal $Cl^-$, efflux of $Cl^-$ facilitates glutamate uptake by generating $\Delta\Psi$. The glutamate/$Cl^-$ exchange mechanism allows an electroneutral SV filling without increasing the osmotic pressure. As the luminal $Cl^-$ concentration gradually decreases, glutamate uptake is driven by the $\Delta\Psi$ generated by the V-ATPase. Glutamate transport finally stops when $\Delta\Psi$ vanishes and luminal pH decreases. **b** In absence of a permeant anion in the vesicular lumen (e.g. $Gluc^-$ or $MeS^-$), glutamate uptake is fuelled solely by the V-ATPase and is limited by the osmolarity. The overall process of loading and acidification is slow. Additionally, if $Gluc^-$ is used as a substitute for $Cl^-$, its buffering capacity prevents luminal acidification in presence of high HEPES concentrations. **c** In absence of VGLUT, acidification is strongly accelerated. KO knockout, $Glu^-$ glutamate, $Gluc^-$ gluconate, $MeS^-$ methanesulfonate. **d** Illustration of glutamate loading at synaptic terminal during recycling of SV. During endocytosis, $Cl^-$ is engulfed inside SV from the extracellular space. The anion is subsequently exchange for glutamate through VGLUT. Left: high $Cl^-$ concentration, right: low $Cl^-$ concentration

**Functional analysis of VGLUT in HEK293T cells**. For electrophysiological analysis of VGLUT anion currents in mammalian cells, we subcloned rat VGLUT1 cDNAs (kindly provided by Dr. Shigeo Takamori) into pcDNA3.1 and mutated the dileucine-like endocytosis motif to alanine as reported by Eriksen et al.[17]. HEK 293 cells passage 3–5 (cell line name; tsA201, Cat. No. 96121229, Sigma-Aldrich) were transiently transfected with 2–5 μg of plasmid DNA using a calcium phosphate precipitation method and examined 36 h later. In brief, DNA was mixed with 330 μl of buffered water (2.5 mM HEPES) and 56.5 μl of 2.5 M $CaCl_2$ by pipetting, subsequently 570 μl of 2× HeBS buffer (280 nM NaCl, 50 mM HEPES, 1.5 mM $Na_2PO_4$, adjusted with 2 M NaOH to pH 7.00) was added dropwise under agitation by vortexing. The DNA-Calcium precipitate was leaved at room temperature for 5 min and thereafter was added to the HEK 293 cells (cultured in a 5 cm dish with 70–80% confluence).

Standard whole-cell patch clamp recordings were performed using an EPC-10 amplifier, controlled by PatchMaster (HEKA, Germany)[58]. Borosilicate pipettes (Harvard Apparatus, USA) were pulled with resistances of 0.9–2 MΩ. Series resistance compensation and capacitance cancellation were applied, resulting in <5 mV voltage error. The standard external solutions contained (in mM) 145 Choline Cl, 50 MES (2-(N-morpholino)ethanesulfonate) 2 Mg-Gluconate, pH 5.0, with and without RB (10 nM). Cells were dialysed with an internal solution containing 10 NaCl, 30 HEPES, 100 TMA Gluconate, 2 $MgCl_2$, 5 EGTA; pH 7.4, and held at −85 mV to established a physiologically low cytoplasmic $[Cl^-]$. The junction potential was calculated and corrected.

**Imaging vesicular $Cl^-$ in living neurons**. For vesicular $Cl^-$ imaging SytClopH was used as a ratiometric indicator by excitation. Imaging was performed at room temperature on an inverted microscope (Axiovert 100TV, Zeiss) equipped with a ×63, 1.2 NA water-immersion objective. Images were acquired with a sCMOS camera (Neo 5.5, Andor) controlled by Andor IQ software (Andor) in 2 × 2 binning mode. SytClopH was excited sequentially at 458, 488 and 565 nm with a computer controlled monochromator (Polychrom V, Till Photonics). Emission was collected with a 525/50 nm Brightline single-band bandpass emission filter when

excited at 458 and 488 nm, and with a 609/54 nm Brightline single-band bandpass emission filter when excited at 565 nm. Calibration of SytClopH was performed on hippocampal cultured neurons[37] using Image J software (US National Institutes of Health) as follows. Intracellular pH and $Cl^-$ were controlled by equilibrating extra- and intracellular ion concentrations for at least 15 min using the $K^+/H^+$ exchanger nigericin (10 μM) and the $Cl^-/OH^-$ exchanger tributyltinchloride (10 μM) in a high $K^+$ containing buffer of different $[Cl^-]$. The buffers of different $[Cl^-]$ were prepared by mixing two solutions containing 4 or 131 mM $Cl^-$. The 4 mM $Cl^-$ solution was composed of 123 mM K gluconate, 2 mM $CaCl_2$, 2 mM $MgSO_4$, 1.2 mM $NaH_2PO_4$, 11 mM glucose, 23 mM HEPES. The 131 mM $Cl^-$ solution contained 123 mM KCl, 2 mM $CaCl_2$, 2 mM $MgCl_2$, 1.2 mM $NaH_2PO_4$, 11 mM glucose, 23 mM HEPES. pH was adjusted with NaOH. The calibrated buffers allowed the measurement of $R_{pH}$ ($F_{488 nm}/F_{458 nm}$) and $R_{Cl}$ ($F_{458 nm}/F_{565 nm}$) at different known $[Cl^-]$ and pH. For all experimental data, pH and $[Cl^-]$ were determined by the following equations[37]:

$$pH = pK_a + \log\left(\frac{R_{pH} - R_A}{R_B - R_{pH}}\right),$$

$$[Cl^-] = K_d^{Cl}\left(\frac{R_{Cl} - R_{free}}{R_{bound} - R_{Cl}}\right),$$

$pK_a$ is the acid dissociation constant of SytClopH. $R_A$ and $R_B$ are the values of $R_{pH}$ for the sensor in its most acidic and basic forms. $R_{free}$ is the maximum value of $R_{Cl}$ when no $Cl^-$ is bound to SytClopH. $R_{bound}$ is the maximum value of $R_{Cl}$ when SytClopH is saturated with $Cl^-$. $R_{bound}$ is proportional to pH with slope $M$, which we determined empirically to be 0.061 per pH unit:

$$R_{bound} = M(pH) + R_{bound, pH0},$$

$K_d^{Cl}$ is the $Cl^-$ dissociation constant which depends on pH:

$$K_d^{Cl} = {}^1K_d^{Cl}\left(\frac{1 + 10^{(pKa-pH)}}{10^{(pKa-pH)}}\right),$$

$^1K_d^{Cl}$ reflects the $Cl^-$ dissociation constant when SytClopH is fully protonated.

Proteolytic cleavage of ClopHensorN from the plasma membrane was performed at room temperature by adding 60 U $ml^{-1}$ AcTEV protease (Sigma-Aldrich) and 1 mM dithiothreitol directly to the living neurons for 15 min[56]. After digestion, cells were washed for 5 min.

**Data analysis**. Fluorescence time series data were analysed with an automated algorithm for bouton detection in order to avoid experimenter bias using self-written macros in Igor Pro (Wavemetrics). For detecting regions of fluorescence intensity changes (functional boutons) the difference image of the images before and at the end of stimulation (maximum change in fluorescence intensity) of a time series was subjected to an à trous wavelet transform with the level $k = 4$ and detection level $l_d = 1.0$[56]. The result is a binary mask showing spots (synaptic boutons) of fluorescence intensity changes that is used to extract individual fluorescence time series of all synaptic boutons in the field of view. For experiments involving VGLUT1 knockout mice we excluded the responses from boutons containing another VGLUT isoform by applying RB at the end of the experiment and sorting the inhibited boutons (725 boutons over a total of 2388; i.e. 30%, in line with a previous study[6]). These VGLUT-positive boutons showed a time constant for acidification similar to boutons from wild-type mice (Supplementary Fig. 9). Only experiments containing >50 active boutons were considered for analysis. All data are represented as mean ± s.e.m. of $n$ experiments. In 'rapid acid quench' experiments, s.e.m. are shown every tenth point (2 s) for better visibility. Acidification time courses were fitted to mono-exponential decay functions using Igor Pro. All statistical tests performed were two-tailed unpaired $t$-tests.

**Estimation of the vesicular pH**. The total fluorescence of SpH in a terminal is the sum of the fluorescence derived from different SpH fractions, including those on the cell surface, those within resting SVs, and those within recycling SVs. We can estimate that only the recycling SVs contribute to the change in fluorescence monitored after the end of the stimulation. Therefore, the fluorescence intensities at each pH can be described by the Henderson–Hasselbalch equation as follows:

$$\Delta F = (F - F_0)/F_{max} = (1/(1 + 10^{nH(pKa-pH)})$$
$$- 1/(1 + 10^{nH(pKa-5.6)}))(1 + 10^{nH(pKa-7.3)}),$$

where $F_0$ is the fluorescence at rest so at a luminal pH of 5.6, $F_{max}$ is the fluorescence at the surface, at pH 7.3, p$Ka$ is the pH at which 50% of the probe is protonated and nH is the Hill coefficient. Upon normalization of the SpH traces between the baseline at rest and the climax of stimulation, $F_0 = 0$ and $F_{max} = 1$. For pHluorin, the p$Ka$ and nH values are 7.09 and 1.35, respectively[47]. From this equation, we can determine the averaged pH of the recycling probes at each

fluorescence intensity $\Delta F$ following:

$$\text{pH} = pKa - 1/nH \log[1/(1/(1 + 10^{nH(pKa-5.6)}) + \Delta F/(1 + 10^{nH(pKa-7.3)})) - 1].$$

The SpH fluorescence decay after stimulation was fitted by a single exponential decay and the vesicular pH reached at steady state is calculated depending on the $\Delta F$ reached at the plateau.

**Data availability**. Data supporting the findings of this manuscript are available from the corresponding authors upon reasonable request.

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

## Acknowledgements

The authors are grateful to K. Tkotz for expert technical assistance and M. Kahms for advices in imaging and valuable discussions. M.M. was the recipient of a Marie Curie Intra-European Fellowship for Career Development (IEF) under the 7th Framework Program (FP7) of the European Commission. J.K. was supported by grants from the DFG (SFB 629, SFB 944 and DFG EXC 1003, Cells in Motion Cluster of Excellence, Münster, Germany) and by a grant of the IZKF Münster (Project No. Kli3/027/15).

## Author contributions

M.M. and J.K. designed the project and wrote the manuscript, M.M. performed experiments on cultured neurons and with J.K. analysed the data, R.E.G. and C.F. planned, performed and analysed experiments on transfected HEK293T cells and provided the *Vglut1*$^{-/-}$ mice.

## Additional information

**Competing interests:** The authors declare no competing financial interests.

