## [Peer Review File · Nature Communications]

Reviewers' comments:

Reviewer #1 (Remarks to the Author):

Martineau and co-workers study acidification and chloride transport of glutamateric synaptic vesicles. Based on the findings that Rose Bengal blocks acidification of WT but not VGLUT-/- synaptic vesicles, and that this blockage depends on the luminal chloride concentration, the authors conclude that VGLUT1 must function as a chloride channel. The authors confirm this by showing a higher chloride concentration in synaptic vesicles from VGLUT-/- neurons utilizing the recently developed chloride sensor ClpHensorN. Although chloride conductance of VGLUT1 has been shown recently in several papers (as acknowledged by the authors), this is still controversial and has only been shown in vitro (purified vesicles/proteins). This study shows chloride conductance in live neurons and I believe this warrants publication in a high-impact journal such as Nature Communications. However, and as I will detail below, I am much less enthusiastic about the second part of the paper (Fig. 6-8) where the authors attempt to prove that VGLUT1 acts as a proton-glutamate antiporter. As I will detail below, I believe the data do not warrant the conclusions drawn by the authors and therefore I cannot recommend publication.

The conclusion that VGLUT1 acts as a proton-glutamate antiporter is based on a 'rapid acid quench' method, where the external medium is acidified resulting in quenching of surface exposed (but not vesicular) pHluorin. The acidification kinetics are then calculated by fitting the acidification kinetics of the remaining vesicular pHluorin. Based on the finding that acidification speed depends on chloride concentration (but not HEPES concentration), the authors conclude that the proton efflux almost equals the size of the proton influx and that the internal buffering capacity must be very small. In line with this, synaptic vesicles in neurons from VGLUT1-/- mice or combining low luminal chloride with Rose Bengal result in supposedly very fast acidification kinetics. I have 2 problems with these data.

First, the curves with Rose Bengal and low chloride (Fig. 7a) have precisely the same shape as the curve with Rose Bengal and high chloride (Fig. 5c). Nevertheless, in the first case, the authors conclude that acidification is accelerated by at least an order of magnitude ("not easily distinguishable from the solution exchange time course during acid quench"; line 228), whereas in the second case the authors interpret the data as no acidification at all. This seems a biased interpretation of the data and the conclusion should be confirmed by other methods. In fact, and this is my second concern, the re-acidification with Rose Bengal and low chloride seems to occur much slower than with high chloride (compare kinetics of Fig 2b with Fig. 1f). This thus directly contradicts the extremely fast acidification observed from the quenching experiments, and I strongly believe the fast acidification observed by the authors is wrong. This directly undermines the conclusion that VGLUT1 acts as a proton/glutamate antiporter and also the quantitative calculations in the final part of the manuscript.

Minor comments:

- The potential buffering by glutamate should be taken into consideration (pKa of 9.5 and 4.1)
- What is the reason for the massive change in acidification upon lowering the concentration of HEPES from 25 to 23.5 mM (Fig. 4b)? Why is it relevant that "The H⁺ buffering capacity of 1.5 mM HEPES corresponds roughly to the expected buffering capacity of 119 mM Gluc" (line 188-189)?

Reviewer #2 (Remarks to the Author):

The mechanisms by which neurotransmitter is pumped into synaptic vesicles following their recycling is one of significant importance and lively debate. For example knowing this mechanism will inform whether this pathway can be modulated, which in turn becomes a potential determinant of synaptic strength. Klingauf and colleagues present in the submitted manuscript a compelling set of data that both the activity of vGlut is required for acidification and the presence of a high Cl concentration which would be a natural consequence of vesicle recycling as the lumen of the vesicle should to first approximation be similar to the extracellular environment (that normally has high Cl). The approach in this paper makes use of a variety of optical tools that can probe the kinetics of vesicle reacidification during recycling and how they depend on vGlut and Cl. On the whole I think this work will make a valuable contribution to the literature on this important topic. The authors however have not written a manuscript that flows particularly well and it makes for some tough reading in parts. Additionally there are numerous technical issues which I feel unnecessarily seed doubt about the data. I am willing to give the authors the benefit of the doubt but future readers may not:

1. The authors chose to normalize the data for most of their optical recordings to the starting fluorescence value. Although this is a common practice for probes that are all confined to the same compartment (say a cytosolic Ca indicator) it is more dangerous when the probe is distributed into different compartments. This largely impacts how one interprets amplitudes of the signals, less so on the kinetics. synaptopHluorin signals are well known to have a large contribution at rest from copies of tagged VAMP2 residing on the axon or bouton surface. Any condition that changes would alter what one calls 100% and therefore mask changes (or lack thereof) in amplitude. I see no reason the authors have chosen to take this path. They could easily normalize their signal to the signal achieved in NH₄Cl superfusion (as they used occasionally). Presenting the data in this fashion would make things much easier to interpret
2. Perhaps the biggest hole in the model derived from this work is the fact that although blocking vGlut with RB dramatically slows acidification of vesicles, the vesicles still acidify in the complete absence of vGlut. The authors build their case that the luminal Cl content is critical in association with vGlut function by showing that they can overcome the acidification block by RB substituting Cl with MES or Gluc. As a control these same experiments should be done in the vGlut KO.
3. The claim that the buffer capacity must be small as change of 1.5 mM HEPES can dramatically slow the reacidification seems an oversimplification. It seems something much more complicated is going on as going from 5 mM HEPES to 23.5 mM has very little impact (and from which one would conclude that SVs have a larger buffer capacity) - thus there is some complex non-linear relationship since going from 23.5 to 25 mM has such a large impact. Interpreting this as evidence for a small buffer capacity seems unwarranted based on this.
4. The authors also note that under normal Cl conditions there is little influence of HEPES on reacidification kinetics as previously reported. As they mention that previous work (Atluri & Ryan) concluded it was either that HEPES did not get in or that it is not a suitable proton buffer for unknown reasons inside vesicles. What these authors did not mention however is that Atluri & Ryan did see a slowing of reacidification with an alternate proton buffer, TRIS with fixed and reasonably high Cl concentration. This too seems counter to the claim of very low endogenous buffer capacity.
5. The acid quenching experiments have a number of hopefully fixable problems:
 - the quench depths seem to be varying a lot under different conditions that do not seem consistent with the direction of the impact on acidification - this may well be an example of the problem of normalizing everything to resting fluorescence (see point 1).
 - the kinetics of acidification are very hard to discern in the graphs (fig 6a). To the unaided eye these lines seem all parallel but with different starting points. Given that in panel b the authors show tau values with highly significant differences I think the authors should show fits of the data so the authors can judge if this is even well describe by a single exponential.
6. The acceleration of acidification (either vGlut KO or RB + Cl removal) is an interesting result. I feel the authors need to make clear however that this is NOT due a dramatic slowing of endocytosis, since that too would lead to what looks like a rapid quench.

Reviewer #3 (Remarks to the Author):

This study analyzes the dynamics of synaptic vesicle (SV) acidification using fluorescence microscopy to infer functional properties of the vesicular glutamate transporter (VGLUT) in live hippocampal neurons. VGLUT1 is the predominant isoform in this model.

A striking finding in the first part of the study is that the pharmacological inhibition of VGLUT1 by Rose Bengal (RB), but not its genetic ablation, impairs re-acidification of endocytosed SVs after glutamate exocytosis. To explain this paradox, the authors postulate that RB inhibits glutamate uptake by VGLUT1, but not its uncoupled anion conductance reported by several groups. Consequently, in wild-type SVs blocked by RB, the high internal concentration of chloride in newly endocytosed vesicles would induce a strong positive polarization that opposes H⁺ pumping by the vesicular V-ATPase, thus delaying re-acidification. This phenomenon would not occur in KO vesicles if VGLUT1 carries the main chloride conductance in the vesicular membrane (the vesicular H⁺/Cl⁻-exchanger CIC3 thus having a minor contribution). In support of this model, substituting impermeant anions for extracellular chloride rescued SV acidification in RB-treated wild-type neurons.

In the second part, the authors compare the dynamics of SV re-acidification under diverse conditions where the V-ATPase is not limited by the electrogenic efflux of chloride through VGLUT1. Under low external chloride, SV-reacidification is much faster in the presence of RB than in its absence, suggesting that H⁺/glutamate antiport by VGLUT1 might delay this acidification. SV re-acidification is also fast in Vglut1 KO neurons.

The authors conclude that among the diverse, and debated, activities of VGLUT reported by in vitro studies, the H⁺/glutamate antiport and the uncoupled chloride conductance are the most significant features under physiological conditions. In the Discussion, they suggest an interesting model in which the refilling of SV with glutamate would successively be driven by two mechanisms. First, in newly endocytosed SVs, glutamate filling would be mainly driven by the positive potential caused by the efflux of internal chloride through VGLUT. In a second phase, when the vesicular chloride gradient becomes shallower, this positive polarization would vanish, increasing H⁺ pumping by the V-ATPase, which becomes the main driving force for glutamate uptake.

The study and the model are interesting. Experiments are rigorously executed. However, many conclusions are supported by very indirect evidence and there are several concerns.

Major points:

1. A major concern is that there is too much interpretation and speculation intermingled with the description of experimental data in the Results section. These parts should be trimmed or moved to the Discussion to let the reader make her/his own mind.
2. The authors postulate that Rose Bengal does not affect the uncoupled anion conductance of VGLUT. This prediction can, and should, be tested using direct assays of VGLUT (for instance, current recording in *Xenopus* oocytes as in ref. 20: Eriksen et al, Neuron 2016).
3. The SytClopH measurements in Fig. 3 suggest that the anion conductance of VGLUT mainly serves for Cl⁻ exit in live neurons, in contrast with its rectification properties (ref. 20) which seem to favor Cl⁻ entry into SVs. It is unclear whether the SytClopH measurements were performed on resting or newly endocytosed vesicles. It would be interesting to compare luminal Cl⁻ levels under these two conditions.
4. The strong increase in SV re-acidification observed when HEPES is decreased from 25 mM to

23.5 mM in the presence of high gluconate (Fig. 4b) is quite surprising and the explanation provided is unclear. Have the authors tested other buffers with distinct capacity?

5. It is argued that in the absence of VGLUT-permeant anions, glutamate uptake is restricted by osmotic pressure (p. 8). This prediction can be tested using, for instance, EPSC recording.

6. The evidence for H⁺/glutamate antiport is very indirect. If the two-phase SV refilling model is correct, inhibiting the V-ATPase in the presence of high chloride should induce an SV alkalinization in newly endocytosed vesicles. This treatment should also partially preserve glutamate refilling and therefore reduce, rather than abolish, mEPSCs.

7. The fast acidification kinetics in Fig. 7 are poorly resolved in time.

8. SVs have been reported to have a large H⁺ buffering capacity in a previous study (Egashira Y et al, J Neurosci. 2015), which should be cited and discussed.

9. Introduction, lines 36-38: the sodium-dependent phosphate transport activity of VGLUTs is not clearly established. The sentence also suggests that the glutamate transport activity is inhibited at the plasma membrane. There is no such statement in the references cited. When targeted to the plasma membrane, VGLUT does transport glutamate under appropriate ionic conditions (Fig. 7 in ref. 20).

Minor points:

a. Red and pink symbols do not contrast easily when used in the same graph.

b. The post-stimulus SV alkalinization discussed on p. 5 is not easily seen in Fig 1f (but clear in Suppl. Fig 4)

c. HEPES concentrations should be indicated on the graphs of Figs 2b and 4a to help notice the reason for the apparent contradiction between the two figures.

d. In Fig. 7 it would be more appropriate to compare the acidification in the presence of RB with controls performed under identical ionic conditions.

Response to the Reviewers

We would like to thank the reviewers for their positive evaluation of our work, raising very valid concerns and making useful suggestions to improve the quality of our study.

Reviewer 1.

Martineau and co-workers study acidification and chloride transport of glutamateric synaptic vesicles. Based on the findings that Rose Bengal blocks acidification of WT but not VGLUT-/- synaptic vesicles, and that this blockage depends on the luminal chloride concentration, the authors conclude that VGLUT1 must function as a chloride channel. The authors confirm this by showing a higher chloride concentration in synaptic vesicles from VGLUT-/- neurons utilizing the recently developed chloride sensor ClopHensorN. Although chloride conductance of VGLUT1 has been shown recently in several papers (as acknowledged by the authors), this is still controversial and has only been shown in vitro (purified vesicles/proteins). This study shows chloride conductance in live neurons and I believe this warrants publication in a high-impact journal such as Nature Communications. However, and as I will detail below, I am much less enthusiastic about the second part of the paper (Fig. 6-8) where the authors attempt to prove that VGLUT1 acts as a proton-glutamate antiporter. As I will detail below, I believe the data do not warrant the conclusions drawn by the authors and therefore I cannot recommend publication.

The conclusion that VGLUT1 acts as a proton-glutamate antiporter is based on a 'rapid acid quench' method, where the external medium is acidified resulting in quenching of surface exposed (but not vesicular) pHluorin. The acidification kinetics are then calculated by fitting the acidification kinetics of the remaining vesicular pHluorin. Based on the finding that acidification speed depends on chloride concentration (but not HEPES concentration), the authors conclude that the proton efflux almost equals the size of the proton influx and that the internal buffering capacity must be very small. In line with this, synaptic vesicles in neurons from VGLUT1-/- mice or combining low luminal chloride with Rose Bengal result in supposedly very fast acidification kinetics. I have 2 problems with these data.

First, the curves with Rose Bengal and low chloride (Fig. 7a) have precisely the same shape as the curve with Rose Bengal and high chloride (Fig. 5c). Nevertheless, in the first case, the authors conclude that acidification is accelerated by at least an order of magnitude ("not easily distinguishable from the solution exchange time course during acid quench"; line 228), whereas in the second case the authors interpret the data as no acidification at all. This seems a biased interpretation of the data and the conclusion should be confirmed by other methods.

We agree that this seems at first glance confusing. However, whether a condition leads to no acidification or rather to very fast reacidification was checked for by the experiments in Fig. 2. Without acid quench, the absence of acidification or of endocytosis would lead to incomplete recovery and a plateau in the SpH response (as for RB and high chloride) while a very fast acidification would lead to fluorescence recovery to baseline (as for RB and low chloride). In addition, we ensure that there is no unquenched, internalized pHluorin (acidification deficit) in VGLUT1 KO synapses by comparing the plateau during the acid quench with and without stimulation, i.e. with and without exo-/endocytosis of synaptic vesicles (orange and grey traces in Fig. 8c). In line with this, we stated in the manuscript: "A deficit in endocytosis and/or acidification in VGLUT1 knockout synapses as alternative explanation could be ruled out. First, the SpH fluorescence fully recovered to baseline (Fig. 2a). Second, the plateaus during the acid quench with and without stimulation, i.e. with and without exo-/endocytosis of synaptic vesicles, were identical, indicating that all vesicles have been re-acidified to steady state (Fig. 8c)."

In fact, and this is my second concern, the re-acidification with Rose Bengal and low chloride seems to occur much slower than with high chloride (compare kinetics of Fig 2b with Fig. 1f). This thus directly contradicts the extremely fast acidification observed from the quenching experiments, and I strongly believe the fast acidification observed by the authors is wrong. This directly undermines the conclusion that VGLUT1 acts as a proton/glutamate antiporter and also the quantitative calculations in the final part of the manuscript.

We agree that at first sight the SpH transients with RB and low chloride appear to recover more slowly than in the control with high chloride. However, the recovery of the curves is composed of the endocytosis and the acidification dynamics. According to Supplementary Fig. 4, RB also induced a transient post-stimulus vesicular alkalinisation (Fig. 1g) caused by exocytosis of V-ATPases that transiently alkalinized cytosolic pH, suggesting that VGLUT inhibition renders SVs sensitive to cytosolic pH changes. Therefore, in Fig. 2b the recovery of the curve of RB and low chloride reflects both, endocytosis and the post-stimulus alkalinisation kinetics. During the acid quench (Fig. 8), the activity of V-ATPase is inhibited at the plasma membrane and this post-stimulus alkalinisation is not observed, leading to the direct measurement of acidification kinetics. To clarify this, we added a sentence in the legend of Figure 2: “Note that glutamatergic SVs are still sensitive to cytosolic pH changes as seen by the remaining transient post-stimulus increase in SpH fluorescence (see Supplementary Fig. 4)”.

Minor comments:

-The potential buffering by glutamate should be taken into consideration (pKa of 9.5 and 4.1)

We changed our calculation taking now into account also the weak but existing proton buffer capacity of glutamate (cf. page 15)

-What is the reason for the massive change in acidification upon lowering the concentration of HEPES from 25 to 23.5 mM (Fig. 4b)? Why is it relevant that “The H⁺ buffering capacity of 1.5 mM HEPES corresponds roughly to the expected buffering capacity of 119 mM Gluc” (line 188-189)?

We chose 23.5 mM HEPES + 119mM Gluconate because they have together the same proton buffering capacity like 25 mM HEPES, i.e. the buffering capacity of 119 mM Gluconate equals that of 1.5 mM HEPES at pH 5.6. We were also puzzled by this large effect in the pHluorin response when lowering HEPES from 25 to 23.5 mM. However, it seems this extremely high non-linearity to a large extent is due to the non-linear pHluorin pH-dependence itself. In other words, if we calculate from the pHluorin signal the pH and plot it against the external buffer concentration most of this non-linearity vanished (newly added Fig. 5e) However, additional experiments with higher HEPES and TRIS concentrations (Supplementary Fig. 8) now suggest that the endogenous buffer capacity roughly equals that of a few tens mM HEPES, in line with a recent theoretical estimate of the luminal buffer capacity (Farsi et al BioEssays 2017). Thus, we had to reconsider our previous conclusion about the very low endogenous buffer capacity and we would like to thank the reviewer for raising this important concern. We changed our manuscript accordingly.

Reviewer 2.

The mechanisms by which neurotransmitter is pumped into synaptic vesicles following their recycling is one of significant importance and lively debate. For example knowing this mechanism will inform whether this pathway can be modulated, which in turn becomes a potential determinant of synaptic strength. Klingauf and colleagues present in the submitted manuscript a compelling set of data that both the activity of vGlut is required for acidification and the presence of a high Cl concentration which would be a natural consequence of vesicle recycling as the lumen of the vesicle should to first approximation be similar to the extracellular environment (that normally has high Cl). The approach in this paper makes use of a variety of optical tools that can probe the kinetics of vesicle reacidification during recycling and how they depend on vGlut and Cl.

On the whole I think this work will make a valuable contribution to the literature on this important topic. The authors however have not written a manuscript that flows particularly well and it makes for some tough reading in parts.

We thank the reviewer for these comments and for judging that our work will make a valuable contribution to the literature. We tried to modify the manuscript and performed additional analyses (Fig. 5) to clarify our conclusions.

Additionally there are numerous technical issues which I feel unnecessarily seed doubt about the data. I am willing to give the authors the benefit of the doubt but future readers may not:

1. The authors chose to normalize the data for most of their optical recordings to the starting fluorescence value. Although this is a common practice for probes that are all confined to the same

compartment (say a cytosolic Ca indicator) it is more dangerous when the probe is distributed into different compartments. This largely impacts how one interprets amplitudes of the signals, less so on the kinetics. synaptophluorin signals are well known to have a large contribution at rest from copies of tagged VAMP2 residing on the axon or bouton surface. Any condition that changes would alter what one calls 100% and therefore mask changes (or lack thereof) in amplitude. I see no reason the authors have chosen to take this path. They could easily normalize their signal to the signal achieved in NH₄Cl superfusion (as they used occasionally). Presenting the data in this fashion would make things much easier to interpret.

As mentioned rightfully by the reviewer, the normalization impacts the amplitudes of the signal but not the kinetics. In this study we are solely interested in the reacidification of those vesicles that are endocytosed in a compensatory fashion triggered by exocytosis (and not at all in the release probability. In the latter case, indeed normalization to the total SV pool assessed by NH₄Cl superfusion would be the method of choice). For our conclusions it is important to quantify the fraction of SVs that do reacidify at all as well as the kinetics of those SVs that show reacidification. For instance, the use of low chloride decreased the excitability of the neurons and thus the release amplitudes, as revealed after normalization to NH₄Cl (Supplementary Fig. 6). Therefore, in order to compare the kinetics of vesicular recovery, we normalized the curves to the intensity at the end of the stimulation, as explained in the text (page 6). In other words, we normalized the curves to the quantity of vesicles released and endocytosed by compensatory endocytosis at a given stimulation.

Finally, we used as much as possible the same set of neurons to test different conditions (before and after application of RB, with different extracellular chloride concentrations, or with and without stimulation during the 'rapid acid quench' strategy, for instance). Thus we aimed at avoiding interference with the luminal pH of resting vesicles in between two tests as much as possible. This idea is reinforced by a recent study which demonstrates that NH₄Cl alters neuronal excitability and synaptic vesicle release (Lazarenko et al., 2017). Since we do not care in this study about the particular release probability and thus signal amplitude, we decided to avoid application of NH₄Cl and its possible effects on the luminal pH of synaptic vesicles.

Lazarenko, R. M., DelBove, C. E., Strothman, C. E. & Zhang, Q. Ammonium chloride alters neuronal excitability and synaptic vesicle release. *Sci. Rep.* **7**, 5061 (2017).

2. Perhaps the biggest hole in the model derived from this work is the fact that although blocking vGlut with RB dramatically slows acidification of vesicles, the vesicles still acidify in the complete absence of vGlut. The authors build their case that the luminal Cl content is critical in association with vGlut function by showing that they can overcome the acidification block by RB substituting Cl with MES or Gluc. As a control these same experiments should be done in the vGlut KO.

This is an interesting suggestion. However, on second thought, it is evident from our results that substituting Cl⁻ with MeS⁻ or Gluc⁻ under RB block as well as ablation of vGlut (KO) work in the same direction, such that Cl⁻ substitution in vGlut KO could experimentally not be distinguished from control (KO only): in both cases, reacidification, as shown by the acid quench (Fig. 8e), takes less than a second. But prompted by the reviewers concern we decided to check directly the presence of a chloride conductance in VGLUT1 and the impact of RB on this conductance using a heterologous system. Therefore, we expressed VGLUT1 in HEK cells and performed whole-cell patch-clamp recordings. As reported previously (Eriksen et al., 2016) we observed a strongly rectifying chloride current (Fig. 3a) which was still present after application of RB (Fig. 3b). In addition to the vesicular Cl⁻ measurement (Fig. 4) our results demonstrate that VGLUT1 transports chloride and this transport is still present after VGLUT1 inhibition by RB.

3. The claim that the buffer capacity must be small as change of 1.5 mM HEPES can dramatically slow the reacidification seems an oversimplification. It seems something much more complicated is going on as going from 5 mM HEPES to 23.5 mM has very little impact (and from which one would conclude that SVs have a larger buffer capacity) - thus there is some complex non-linear relationship since going from 23.5 to 25 mM has such a large impact. Interpreting this as evidence for a small buffer capacity seems unwarranted based on this.

We thank the reviewer for the valid concern. To better understand the impact of HEPES concentration on the vesicular acidification, we performed further analyses and converted the final fluorescence intensity reached after vesicular recycling in presence of different Cl⁻ and HEPES concentrations into

pH values (Figure 5c and 5e). With these analyses, we overcome the non-linear pH-dependence of pHluorin and it appears that lowering the concentration of HEPES from 25 to 5 mM has a progressing impact on the final vesicular pH reached. Therefore, we had to reconsider our previous conclusion about the small endogenous buffer capacity. Additional experiments with higher HEPES and TRIS concentrations (Supplementary Fig. 8) now suggest that the endogenous buffer capacity roughly equals that of a few tens mM HEPES, in line with a recent theoretical estimate of the luminal buffer capacity (Farsi et al., 2017). We added these new experiments (Supplementary Fig. 8) and changed our conclusions accordingly.

4. The authors also note that under normal Cl conditions there is little influence of HEPES on reacidification kinetics as previously reported. As they mention that previous work (Atluri & Ryan) concluded it was either that HEPES did not get in or that it is not a suitable proton buffer for unknown reasons inside vesicles. What these authors did not mention however is that Atluri & Ryan did see a slowing of reacidification with an alternate proton buffer, TRIS with fixed and reasonably high Cl concentration. This too seems counter to the claim of very low endogenous buffer capacity.

As already mentioned in our response to the previous comment, we studied the impact of TRIS as an alternate proton buffer on the reacidification kinetics (Supplementary Fig. 8). We found that 40 mM TRIS as well as 40 mM HEPES had a significant effect on the acidification kinetics compared to 5 mM of buffer. In line with TRIS having a higher pKa compared to HEPES (8.3 and 7.55 respectively), raising TRIS concentration has a significantly lesser effect compared to the same increase in HEPES concentration. Yet at 5 mM TRIS has a higher potency compared to HEPES, which is consistent with previous observations (Atluri and Ryan, 2006). Like Atluri and Ryan we have no explanation for this anomaly at 5 mM.

5. The acid quenching experiments have a number of hopefully fixable problems:

- the quench depths seem to be varying a lot under different conditions that do not seem consistent with the direction of the impact on acidification - this may well be an example of the problem of normalizing everything to resting fluorescence (see point 1).

Yes, this is a fixable problem. Normalization to ammonium chloride would not help here, but with normalization between 0 and 1 this information would indeed be lost. With our normalization both, the rate of acidification and the pH reached (quench depth) are independent and can be distinguished. For instance, at low chloride the rates of acidification are similarly slow independent of the respective Cl substitute or the HEPES concentration used (Fig. 7), while the final pH values reached are different (Fig. 5). As a consequence, the quench depths and the plateaus reached during the quench depend on the final pH reached but not on the rate of acidification (Fig. 7).

- the kinetics of acidification are very hard to discern in the graphs (fig 6a). To the unaided eye these lines seem all parallel but with different starting points. Given that in panel b the authors show tau values with highly significant differences I think the authors should show fits of the data so the authors can judge if this is even well describe by a single exponential.

We added a new panel b to Fig. 6 (now Fig. 7) in order to show a higher magnification of the curves during the acid quench with single exponential fits.

6. The acceleration of acidification (either vGlut KO or RB + Cl removal) is an interesting result. I feel the authors need to make clear however that this is NOT due a dramatic slowing of endocytosis, since that too would lead to what looks like a rapid quench.

We agree with the reviewer. However, in order to correctly interpret the acid quench results and to not mistake a rapid quench for a dramatic slowing of endocytosis, we checked also for endocytosis deficits for all these conditions using a standard pHluorin exocytosis/endocytosis paradigm (Figs 2 and 5). Actually, we explained this in the manuscript: "A deficit in endocytosis and/or acidification in VGLUT1 knockout synapses as alternative explanation could be ruled out. First, the SpH fluorescence fully recovered to baseline (Fig. 2a). Second, the plateaus during the acid quench with and without stimulation, i.e. with and without exo-/endocytosis of synaptic vesicles, were identical indicating that all vesicles have been re-acidified to steady state (Fig. 8c)."

Reviewer 3.

This study analyzes the dynamics of synaptic vesicle (SV) acidification using fluorescence microscopy to infer functional properties of the vesicular glutamate transporter (VGLUT) in live hippocampal neurons. VGLUT1 is the predominant isoform in this model.

A striking finding in the first part of the study is that the pharmacological inhibition of VGLUT1 by Rose Bengal (RB), but not its genetic ablation, impairs re-acidification of endocytosed SVs after glutamate exocytosis. To explain this paradox, the authors postulate that RB inhibits glutamate uptake by VGLUT1, but not its uncoupled anion conductance reported by several groups. Consequently, in wild-type SVs blocked by RB, the high internal concentration of chloride in newly endocytosed vesicles would induce a strong positive polarization that opposes H⁺ pumping by the vesicular V-ATPase, thus delaying re-acidification. This phenomenon would not occur in KO vesicles if VGLUT1 carries the main chloride conductance in the vesicular membrane (the vesicular H⁺/Cl⁻ exchanger CIC3 thus having a minor contribution). In support of this model, substituting impermeant anions for extracellular chloride rescued SV acidification in RB-treated wild-type neurons.

In the second part, the authors compare the dynamics of SV re-acidification under diverse conditions where the V-ATPase is not limited by the electrogenic efflux of chloride through VGLUT1. Under low external chloride, SV-reacidification is much faster in the presence of RB than in its absence, suggesting that H⁺/glutamate antiport by VGLUT1 might delay this acidification. SV re-acidification is also fast in Vglut1 KO neurons.

The authors conclude that among the diverse, and debated, activities of VGLUT reported by in vitro studies, the H⁺/glutamate antiport and the uncoupled chloride conductance are the most significant features under physiological conditions. In the Discussion, they suggest an interesting model in which the refilling of SV with glutamate would successively be driven by two mechanisms. First, in newly endocytosed SVs, glutamate filling would be mainly driven by the positive potential caused by the efflux of internal chloride through VGLUT. In a second phase, when the vesicular chloride gradient becomes shallower, this positive polarization would vanish, increasing H⁺ pumping by the V-ATPase, which becomes the main driving force for glutamate uptake.

The study and the model are interesting. Experiments are rigorously executed. However, many conclusions are supported by very indirect evidence and there are several concerns.

We thank the reviewer for his/her enthusiasm with respect to our study and for appreciating our experiments as rigorously executed.

Major points:

1. A major concern is that there is too much interpretation and speculation intermingled with the description of experimental data in the Results section. These parts should be trimmed or moved to the Discussion to let the reader make her/his own mind.

We took into account the comment of the reviewer while correcting the manuscript. Notably, the estimations of the proton flux and transport rate have now been moved to the discussion.

*2. The authors postulate that Rose Bengal does not affect the uncoupled anion conductance of VGLUT. This prediction can, and should, be tested using direct assays of VGLUT (for instance, current recording in *Xenopus* oocytes as in ref. 20: Eriksen et al, *Neuron* 2016).*

We thank the reviewer for this interesting suggestion. To directly demonstrate the existence of a chloride conductance in the presence of RB we heterologously expressed VGLUT1 in HEK cells and performed whole-cell patch-clamp recordings (newly added Fig. 3). As reported previously (Eriksen et al., 2016) we observed a strongly rectifying chloride current (Fig. 3a) which was still present after application of Rose Bengal (Fig. 3b).

3. The SytClopH measurements in Fig. 3 suggest that the anion conductance of VGLUT mainly serves for Cl⁻ exit in live neurons, in contrast with its rectification properties (ref. 20) which seem to favor Cl⁻ entry into SVs. It is unclear whether the SytClopH measurements were performed on resting or newly endocytosed vesicles. It would be interesting to compare luminal Cl⁻ levels under these two conditions.

In order to detect the fluorescence from a pure vesicular population with reasonable signal to noise ratio, the SytClopH probes present on the cell surface have been cleaved by a TEV protease and the results presented in Fig. 4 have been performed on resting neurons. Of course, we also performed experiments on recycling vesicles (without cleavage) during stimulation, but we had to face technical limitations. First, during vesicular recycling, the pH changes from 7.3 to 5.6 while at the same time the Cl⁻ concentration varies from about 130 to 15 mM. Therefore, they reach values at the limit of the sensor sensitivity (Fig. 4b and 4c) leading to high variability in the signal. Second, the signal originates from different probe fractions: cell surface, immobile resting vesicles, and endocytosed re-acidifying vesicles. Since all vesicles are not endocytosed at the same time the contribution of each fraction to the total fluorescence changes at each time point. Finally, the time-lapse recordings induced a different bleaching rate in the three channels leading to changes in the ratio even at rest. In conclusion, due to these limitations all inducing variability of the signal, we refrained from including these data, which cannot unequivocally be interpreted.

4. The strong increase in SV re-acidification observed when HEPES is decreased from 25 mM to 23.5 mM in the presence of high gluconate (Fig. 4b) is quite surprising and the explanation provided is unclear. Have the authors tested other buffers with distinct capacity?

We thank the reviewer for this valid concern. We were also puzzled by the unexpected strong non-linearity. To clarify this part, the pH dependence of the pHluorin fluorescence was calibrated (Sankaranarayanan et al., 2000 and our own data), thus allowing the conversion of the final fluorescence intensity reached after vesicular recycling into pH values (Fig. 5c, 5e and Methods). It seems, this extremely high non-linearity to a large extent is merely due to the non-linear pHluorin pH-dependence itself. In addition, we now tested the influence of an alternative buffer, TRIS on the vesicular reacidification kinetics (Supplementary Fig. 8). Our results now suggest that the endogenous buffer capacity roughly equals that of a few tens mM HEPES, in line with a recent theoretical estimate of the luminal buffer capacity (Farsi et al., 2017).

5. It is argued that in the absence of VGLUT-permeant anions, glutamate uptake is restricted by osmotic pressure (p. 8). This prediction can be tested using, for instance, EPSC recording.

We agree with the reviewer that measuring directly the concentration of glutamate in synaptic vesicles would be a great addition to the study – and we attempted it. However, here we faced technical limitations. We manipulated the extracellular chloride concentration shortly before the beginning of the measurement to minimize a coupling to the cytosolic anion concentrations. While our pHluorin measurements probe the newly endocytosed synaptic vesicles, postsynaptic EPSCs reflect the glutamate content of exocytosed vesicles drawn from a large pool of SVs. Therefore, a large fraction of the vesicular pool needs to be turned over in presence of low chloride concentrations (which decreases strongly neuronal excitability) before monitoring their release and thus the EPSCs during a second round of exocytosis. Depleting a significant fraction of vesicles in these conditions is tricky and most importantly we then could not distinguish between the luminal and the cytosolic chloride concentration change. This is important to consider because it has been shown that the cytosolic chloride concentration influences VGLUT activity (for a review on extravesicular chloride effect on VGLUT see Omote et al., 2011). Even with bafilomycin it took us a long time and required a lot repeated stimulation to detect a small decrease in miniature EPSC amplitude, fully in line with a previous publication (Zhou et al., 2000). Alternatively, we thought about using a low affinity glutamate sensor. However, to our knowledge, the variant with the lowest affinity has a K_d of about 1 mM (Okumoto et al., 2005), and thus will be saturated at the expected luminal glutamate concentration of 100 to 150 mM.

Okumoto, S., Looger, L. L., Micheva, K. D., Reimer, R. J., Smith, S. J. & Frommer, W. B. Detection of glutamate release from neurons by genetically encoded surface-displayed FRET nanosensors. *Proc. Natl. Acad. Sci. USA* **102**, 8740-8745 (2005).

6. The evidence for H⁺/glutamate antiport is very indirect. If the two-phase SV refilling model is correct, inhibiting the V-ATPase in the presence of high chloride should induce an SV alkalinization in newly endocytosed vesicles. This treatment should also partially preserve glutamate refilling and therefore reduce, rather than abolish, mEPSCs.

We observed that VGLUT acts as a proton shunt (Figs. 7 and 8), leading us to propose that VGLUT is a glutamate/H⁺ exchanger. After inhibition of the V-ATPase with Folimycin an alkalinization is sometimes observed at individual synaptic boutons (difficult to distinguish from a proton leak from resting vesicles), but usually is not obvious anymore in the average trace for many boutons (Supplementary Fig. 4). However, an alkalinization could only be detected, if free protons were present or if the affinity of VGLUT for proton binding was higher than that of HEPES or the luminal protein matrix. VGLUT has a low affinity for glutamate (K_m = 1-3 mM) and we thus can expect a similar affinity for protons. Given the pK_a of HEPES of 7.55 and that there are no free protons in the vesicle lumen (pH 7.4 corresponds to about 0.5 ‰ of a free proton), it is actually not surprising to observe no transport and no alkalinization when the V-ATPase is inhibited. $\Delta\Psi_{Cl}$ is the driver for glutamate/proton exchange by VGLUT, but in order to have glutamate uptake, protons need to be present in the lumen of the vesicle or have to be provided by V-ATPase activity. Interestingly, in line with this, the group of Takamori has shown that even without ATP, VGLUT can load glutamate into large liposomes solely when Cl is present in the lumen (Fig. 3b of Schenck et al., 2009).

7. The fast acidification kinetics in Fig. 7 are poorly resolved in time.

We added higher magnification versions of the curves during the acid quench with single exponential fits.

8. SVs have been reported to have a large H⁺ buffering capacity in a previous study (Egashira Y et al, J Neurosci. 2015), which should be cited and discussed.

We added a remark in our discussion. Egashira et al aimed at determining the vesicular proton buffer capacity by titration with different ammonium concentrations. This should well work in e.g. liposomes, i.e. in systems in equilibrium, where no other fluxes are active, but in our opinion not in SVs: they used a simple buffer equation, assuming only mass balance for NH₃, NH₄⁺ and protons inside and outside the vesicle, and in this way arrived at a significantly higher estimate for the endogenous buffer capacity of 57 mM NH₄⁺/pH. However, this simple model neglects all the other possible ion fluxes (which we show in our paper) in SVs induced during titration, like by vATPase activity triggered by pH neutralization, and thus should lead to an overestimation of the buffer capacity.

9. Introduction, lines 36-38: the sodium-dependent phosphate transport activity of VGLUTs is not clearly established. The sentence also suggests that the glutamate transport activity is inhibited at the plasma membrane. There is no such statement in the references cited. When targeted to the plasma membrane, VGLUT does transport glutamate under appropriate ionic conditions (Fig. 7 in ref. 20).

We thank the reviewer for noticing this error – a remnant of an early version of the manuscript. We now modified the sentence to be correct and more accurate.

Minor points:

a. Red and pink symbols do not contrast easily when used in the same graph.

We modified the colors in the graphs of Fig. 1.

b. The post-stimulus SV alkalinization discussed on p. 5 is not easily seen in Fig 1f (but clear in Suppl. Fig 4)

We added the panel g in Fig. 1 in order to show the post-stimulus SV alkalinization more clearly.

c. HEPES concentrations should be indicated on the graphs of Figs 2b and 4a to help notice the reason for the apparent contradiction between the two figures.

The HEPES concentrations were noticed in the legends of Fig. 2 and Fig. 4 (now Fig. 5). They now appear also on the graphs.

d. In Fig. 7 it would be more appropriate to compare the acidification in the presence of RB with controls performed under identical ionic conditions.

We modified Fig. 7 (now Fig. 8) accordingly. We now compare the acidification in the presence of RB with controls performed under identical ionic conditions. For clarity, we separated results with Gluconate (Fig. 8a) from results with Methanesulfonate (Fig. 8b). As a consequence, we also modified the panel 8e.

Reviewers' Comments:

Reviewer #1:

Remarks to the Author:

The authors added new explanations and data that sufficiently address all my comments. Overall, I think the manuscript improved much (although it is still a very complex read). This study reveals an important new mechanism of neurotransmitter loading into synaptic vesicles (proton-glutamate exchange) and confirms the controversially discussed chloride conductance of VGLUT. I recommend publication.

Reviewer #2:

Remarks to the Author:

The authors have addressed all my concerns. I think this is an important paper and they have shed important new light on the mechanism of neurotransmitter filling. In my opinion this is certainly ready for publications.

The following mistakes in the figure legends were found:

- 1) Legend of Figure 1: the explanation for panel d and e are crossed.
- 2) Fig 3 Line 725  "HEK cells" not just cells.

Reviewer #3:

Remarks to the Author:

The concerns raised have been satisfactorily addressed and the manuscript is significantly improved.

However, there are still a few places where conclusions are a bit overstated.

For instance, the conclusion that data in Fig. 5 'demonstrate that Cl⁻ efflux controls the amount of glutamate loaded into SVs' (Results, lines 200-202) seems too assertive for experiments exclusively measuring SV reacidification, not glutamate loading.

Similarly, I agree with the authors that the effect of gluconate in Fig. 5a probably reflects an osmotic constraint (lines 188-192), the uncoupled anion conductance of VGLUT being impermeant to gluconate. However, there is no direct evidence for this osmotic balance effect.

The abstract should make it clear by redrafting the sentence on line 23:

'... replaced by glutamate in an electrically, and presumably osmotically, neutral manner.'

Otherwise this is a nice study which provides important information about the role of VGLUT in SV recycling.

Response to the Reviewers

Reviewer #1.

The authors added new explanations and data that sufficiently address all my comments. Overall, I think the manuscript improved much (although it is still a very complex read). This study reveals an important new mechanism of neurotransmitter loading into synaptic vesicles (proton-glutamate exchange) and confirms the controversially discussed chloride conductance of VGLUT. I recommend publication.

We thank the reviewer for his/her enthusiasm with respect to our study.

Reviewer #2.

The authors have addressed all my concerns. I think this is an important paper and they have shed important new light on the mechanism of neurotransmitter filling. In my opinion this is certainly ready for publications.

We thank the reviewer for these comments.

The following mistakes in the figure legends were found:

1) Legend of Figure 1: the explanation for panel d and e are crossed.

We corrected these mistakes.

2) Fig 3 Line 725  "HEK cells" not just cells.

We modified the text accordingly.

Reviewer #3.

The concerns raised have been satisfactorily addressed and the manuscript is significantly improved. However, there are still a few places where conclusions are a bit overstated.

For instance, the conclusion that data in Fig. 5 demonstrate that Cl⁻ efflux controls the amount of glutamate loaded into SVs (Results, lines 200-202) seems too assertive for experiments exclusively measuring SV reacidification, not glutamate loading.

Similarly, I agree with the authors that the effect of gluconate in Fig. 5a probably reflects an osmotic constraint (lines 188-192), the uncoupled anion conductance of VGLUT being impermeant to gluconate. However, there is no direct evidence for this osmotic balance effect.

The abstract should make it clear by redrafting the sentence on line 23: replaced by glutamate in an electrically, and presumably osmotically, neutral manner.

Otherwise this is a nice study which provides important information about the role of VGLUT in SV recycling.

We thank the reviewer for his/her positive evaluation. We hope that we now found all places where we overenthusiastically somewhat overstated the conclusions. Likewise, we modified the sentence in the abstract as advised.